# Force Prompting: Video Generation Models Can Learn and Generalize Physics-based Control Signals

**Nate Gillman**
Brown University

**Charles Herrmann**[*]
Google DeepMind

**Michael Freeman**
Brown University

**Daksh Aggarwal**
Brown University

**Evan Luo**
Brown University

**Deqing Sun**
Google DeepMind

**Chen Sun**[*]
Brown University

## Abstract

Recent advances in video generation models have sparked interest in world models capable of simulating realistic environments. While navigation has been well-explored, physically meaningful interactions that mimic real-world forces remain largely understudied. In this work, we investigate using physical forces as a control signal for video generation and propose *force prompts* which enable users to interact with images through both localized point forces, such as poking a plant, and global wind force fields, such as wind blowing on fabric. We demonstrate that these force prompts can enable videos to respond realistically to physical control signals by leveraging the visual and motion prior in the original pretrained model, without using any 3D asset or physics simulator at inference. The primary challenge of force prompting is the difficulty in obtaining high quality paired force-video training data, both in the real world due to the difficulty of obtaining force signals, and in synthetic data due to limitations in the visual quality and domain diversity of physics simulators. Our key finding is that video generation models can *generalize* remarkably well when adapted to follow physical force conditioning from videos synthesized by Blender, even with limited demonstrations of few objects (e.g., flying flags, rolling balls). Our method can generate videos which simulate forces across diverse geometries, settings, and materials. We also try to understand the source of this generalization and perform ablations on the training data that reveal two key elements: visual diversity and the use of specific text keywords during training. Our approach is trained on only around 15k training examples for a single day on four A100 GPUs, and outperforms existing methods on force adherence and physics realism, bringing world models closer to real-world physics interactions. We release all datasets, code, model weights, and interactive video demos at our project page, https://force-prompting.github.io/.

## 1 Introduction

Humans develop an intuitive understanding of how objects respond to forces since infancy (Wilkening and Cacchione, 2010; Ullman et al., 2017): a gentle poke causes a plant to sway, while a breeze creates rippling patterns across fabric. Do video generation models, which encode powerful visual and motion priors through internet-scale pretraining, possess a similar level of intuitive physics understanding? And if so, how to elicit their capabilities to interact with force inputs? A positive answer to these questions would provide a more flexible and expressive interface for video content creation, enable interactive video generation with user input (e.g., generating a video game), and eventually lead to an intuitive world model for intelligent agents to plan and make decisions with.

---

[*]Equal advising. Correspondence to: nate_gillman@brown.edu, chensun@brown.edu. 39th Conference on Neural Information Processing Systems (NeurIPS 2025).

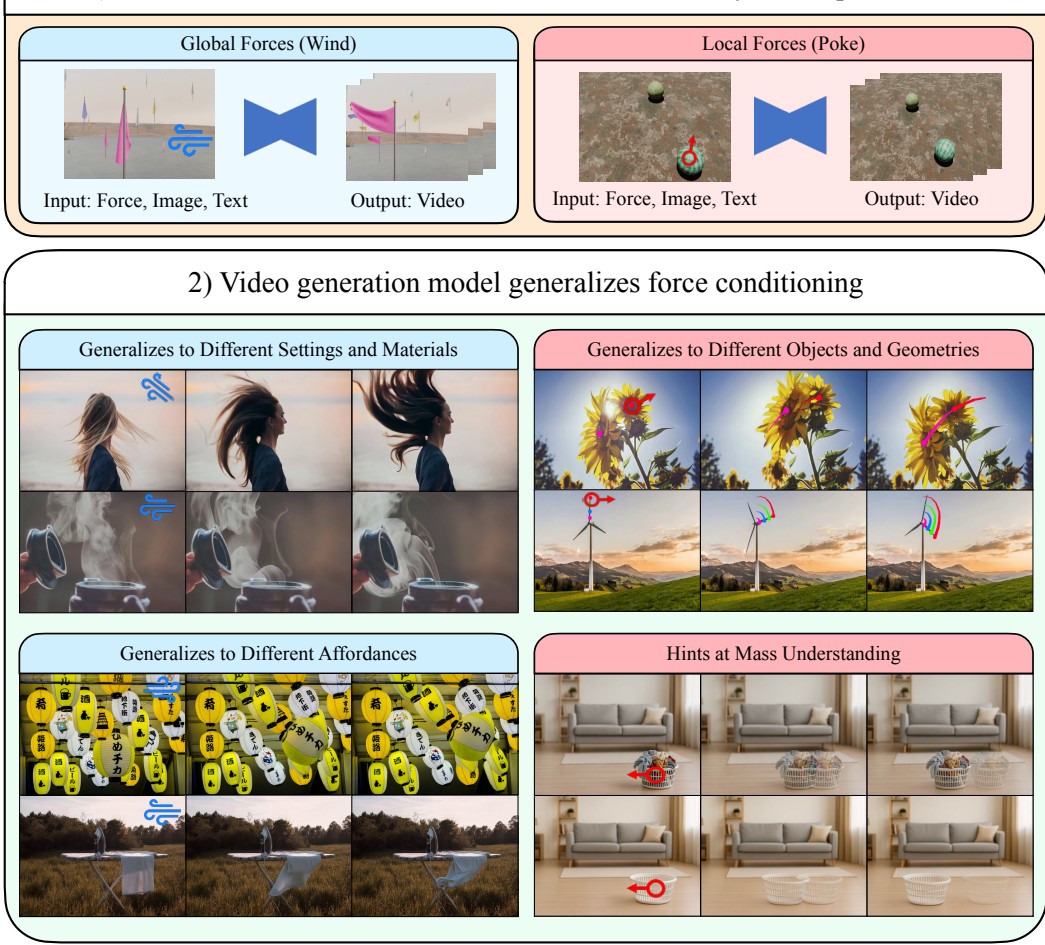

Figure 1: **Force prompting** allows users to apply either global or local forces to objects in an image and then generate the resultant video. Despite being trained on a limited set of synthetic videos (15k for global force and 23k for local force), we observe significant generalization to different settings, materials, objects, geometries, affordances, and some initial hints at mass understanding. Trajectory visualization or alpha overlay are incorporated to better illustrate movement for some examples.

We introduce **Force Prompting**, a step towards incorporating force-based control (direction and magnitude) into video generation models. We explore two distinct categories of force prompts: *local* force prompts, such as instantaneous pokes or pulls applied to specific regions, and *global* force prompts, such as sustained directional wind that affects the entire scene uniformly. Crucially, as manually collecting force annotations from natural videos is both costly and difficult, we instead leverage physics simulators (e.g., Blender) to hand-craft perfectly annotated training data. With our data creation pipeline, we specify a collection of objects along with the force conditions, and simulate the resulting dynamics to obtain the paired training videos. We hypothesize that such *sim2real* generalization is feasible because state-of-the-art video generation models already encode strong priors about visual dynamics, and our paired force-video data serves the role of eliciting their understanding of the physics-based control signals.

We implement Force Prompting by introducing additional force control as local or global vector fields on a video generation model (Yang et al., 2024) conditioned on initial frame and text. We also curate an evaluation benchmark of diverse objects and motion types to evaluate global and local force prompts. As illustrated in Figure 1, our main finding is that despite the synthetic visual appearance and few objects (flying flags and rolling balls) in our training data, video generation models can indeed learn to execute fine-grained force prompts, and exhibit surprisingly strong *generalization* behavior across diverse settings, object shapes and materials, geometry, and affordances. Through

extensive human evaluations, we demonstrate that Force Prompting exhibits superior adherence to physical instruction while maintaining realistic motion and visual quality, when compared to text-conditioned baselines. This validates our hypothesis that synthetic data can teach video generation models intuitive physics and control without damaging their video priors. We further show that simply extrapolating the future by treating forces as local trajectories is insufficient, and our approach significantly outperforms the state-of-the-art in trajectory-controlled video generation (Geng et al., 2024). Notably, Force Prompting can be trained in approximately a single day on four NVIDIA A100 GPUs. We also try to understand the cause of this strong generalization and perform a careful ablation on the training data. We find two elements that appear important to generalization: visual diversity in the training data with respect to the control signal and the usage of certain text keywords at training time, which appear to help elicit the understanding of force control signals.

In summary, our main contributions are as follows:

1. We introduce physical forces as conditioning signals for video generation through two models: one for localized point forces and another for global wind forces.

2. We find that video models can execute precise force prompts with broad generalization to different settings, objects, geometries, and affordances despite minimal training data (15K videos) and modest computational resources (one day on four A100 GPUs). We also attempt to understand the source of this generalization and perform careful ablations on the training data, finding two key elements: visual diversity with respect to the control signal, as well as the usage of text keywords at training time, which appear to help elicit understanding of force control signals.

3. We show that our force-conditioned model has some degree of mass understanding, where the same force can cause a lighter object to move farther than a heavier one.

We release all datasets, code, and models on our project page, https://force-prompting.github.io/.

## 2  Related Works

**Video generation:** In the last several years, video generation models have made rapid progress in visual quality and realistic dynamics (Singer et al., 2022; Ho et al., 2022; Blattmann et al., 2023; Girdhar et al., 2023; Bar-Tal et al., 2024; Brooks et al., 2024). In particular, Sora (Brooks et al., 2024) was one of the first video generation models to demonstrate truly compelling diverse real-world physical phenomena and directly advocated for the future use of using video generators as simulators for the physical world. In the last half year, significant progress has been made by open source models such as CogVideoX (Yang et al., 2024) and Wan 2.1 (Wang et al., 2025), even approaching the quality of closed-source models. While these models act as strong video priors, they primarily use text and images as input and lack precise control over general actions or other physical inputs.

**Controllable video generation:** As video models have rapidly progressed, so too has the accompanying field of controllability for these models with the majority of work in this domain focusing on either camera control (He et al., 2024; Zheng et al., 2024; Sun et al., 2024) or various paradigms of motion control (Yin et al., 2023; Chen et al., 2023; Wang et al., 2024; Shi et al., 2024; Niu et al., 2024; Wu et al., 2024; Geng et al., 2024; Li et al., 2024a; Namekata et al., 2024; Zhang et al., 2024b), such as drag-based, trajectory-based, and optical flow-based techniques. Many of the existing motion control models Yin et al. (2023); Chen et al. (2023); Zhang et al. (2024b) require the complete pre-specified trajectory, specifying the location of the pixel on every generated frame. This reliance on full temporal information makes it difficult to use these models for simulation or prediction tasks.

Motion Prompting (Geng et al., 2024), a concurrent work, uses spatio-temporally sparse trajectories as a conditioning signal, enabling users to specify motion over a few frames for video extrapolation. While this might superficially resemble force control, crucial distinctions exist. First, global phenomena like wind or fluid dynamics are naturally expressed as forces but are difficult or impossible to represent with trajectories. Second, applied forces fundamentally depend on an object's mass or material properties - a dependency absent when specifying motion or location (e.g., identical forces induce greater displacement in lighter objects). Third, specifying an object's location across a few frames is not equivalent to an applied force; the same observed motion could result from numerous alternative causes, such as camera movement or internal object changes. We compare to Motion Prompting and demonstrate significantly better adherence to the conditioning force.

**Interactive world models:** Paralleling the interest in video generation models, interactive world models (Ha and Schmidhuber, 2018) have gained significant attention. Despite extensive research in this area, investigations have predominantly concentrated on video game environments Valevski et al. (2024); Che et al. (2024); Bruce et al. (2024). While a few contemporary studies have begun exploring real-world applications (e.g., Bar et al. (2024); Agarwal et al. (2025)), none explores interactions besides camera control or text. In contrast, our work focuses on interaction through physical forces.

**Physical simulators and hybrid approaches:** Early work (Davis et al., 2015a,b) on generating video based on intuitive forces extracts modal bases of vibrating objects in 2D image space; these works, as well as their modern adaptations (Li et al., 2024b), represented motion as a series of vibrations with different frequencies and intensities, which works well for vibration-like motions but struggles to represent many types of motion, such as linear motion. This led to an alternative research direction explicitly incorporating physics solvers (Chen et al., 2022; Zhong et al., 2024; Le Cleac'h et al., 2023; Xie et al., 2024; Zhang et al., 2024a; Huang et al., 2024; Liu et al., 2024a; Lin et al., 2024; Aira et al., 2024). However, almost all of these techniques require the 3D geometry of the scenes. Recent work has focused on combining both physics simulators and generative models, trying to get the best of both worlds: accurate dynamics from the simulator and better appearance from generative models. For example, PhysGen (Liu et al., 2024b) uses a rigid-body physics solver to model object collisions and then renders these scenes through a video generator, and PhysMotion (Tan et al., 2024) uses a combination of a 3D physics solver and a video generation model. However, due to their usage of physics simulators, they are limited in the types of dynamics they can model. In contrast, we explore using the video generation as a simulator and do not use a physics simulator at inference time. We mention some concurrent works as well: (Li et al., 2025a) also explores the use of simulated videos to finetune generative models, but their focus is on modeling object freefall as opposed to learning physics-based control; Li et al. (2025b) explores action-conditioned video generation, but their model requires the use of a physics simulator at inference time; and Wang* et al. (2025) explores force-conditioned video generation, but their framework requires learning a 3D point cloud trajectory model from synthetic data, and then passing that 4D temporal volume into a point cloud-conditioned video generation model.

# 3 Method: Force Prompting

The goal of Force Prompting is to enable users to interact with images through physical forces. To this end, we explore two distinct force prompts paradigms: a global model that allows users to animate an entire scene with directional wind forces, and a local model that enables precise interaction through localized point forces applied to specific objects within the image. Our video generation method takes as input a triple $(\tau, \phi, \pi)$, where $\tau$ is the text prompt, $\phi \in \mathbb{R}^{c \times h \times w}$ is the initial frame with height $h$ width $w$ and $c$ channels, and $\pi$ is the physics control signal which represents the force being applied: for the wind force model, this is simply a force vector (magnitude, angle) $\in \mathbb{R}^2$, and for the point force model, this is a force vector (magnitude, angle) $\in \mathbb{R}^2$ along with pixel coordinates $(x, y) \in \mathbb{R}^2$ specifying where to apply the force. The goal is to generate a video $v \in \mathbb{R}^{f \times c \times h \times w}$. While we train the global force and local force models with different synthetic datasets and encode the force inputs differently for each, both models share identical architectures and training procedures.

## 3.1 Synthetic training data

To construct our global wind force dataset, we use a physics simulator to generate videos of flags waving in the wind. And we construct our local point force dataset in two parts: for the first part, we use a physics simulator to generate videos of a ball rolling across the ground; and for the second part, we use a model (Zhang et al., 2024a) which integrates 3D Gaussians and a physics simulator to generate videos of a plant being poked. We provide more detail below.

**Global force dataset**: We use Blender to construct a dataset of flags waving in response to varying wind conditions. In order to generate a diverse dataset, we randomize multiple parameters for each video: flag quantity ($\mathrm{Unif}\{1, \ldots, 64\}$), flag color (from a set of 100), flag positions, camera placement, HDRIs (High Dynamic Range Images), which are 360-degree panoramic images used for lighting and background purposes (selected from 50 options on Polyhaven), wind direction in $[0, 360)$, and wind speed in $[0, 1]$, where 0 corresponds to no wind, and 1 corresponds to very strong wind. Each video captures the flags' transitions from stationary to wind-affected state. Our training dataset has 15k videos.

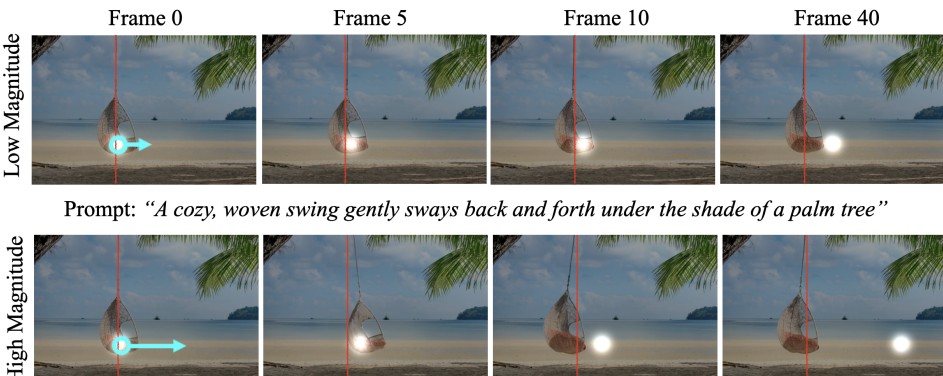

Figure 2: **Visualizing the point force control signal.** The magnitude of applied force is proportional to the gaussian blob's velocity in the control signal, producing proportionally stronger impulses. Stronger forces (bottom) generate faster-moving blobs and correspondingly larger physical responses than gentler forces (top). Note, red line added at the same location in each image for visualization. In our method, we enable the force prompt to dictate the object's trajectory, deliberately excluding such specifics from the text prompt.

**Local force dataset**: The first scenario in our dataset comprises 12k videos of balls, with one of them rolling in response to being pushed by an unseen point-wise force (the force actor is *not rendered*), and the other balls remaining stationary. We generate these videos using Blender with randomized parameters: ball quantity ($\mathrm{Unif}\{2, 3, 4\}$), ball textures (soccer balls using a Polyhaven mesh [$p = 2/3$] , or bowling balls modeled as smooth spheres [$p = 1/3$]), ball colors (from a set of 108), ball positions, camera position, ground textures (from 42 Polyhaven options), target ball selection, force angle in $[0, 360)$, and force magnitude in [0,1]. We assign the bowling ball to be four times the mass of the soccer ball with the goal of teaching the model mass-based dynamics. The second scenario (11k videos) utilizes PhysDreamer (Zhang et al., 2024a), a *generative-simulator hybrid*, and features videos of a carnation swaying back and forth in response to being poked by an unseen force. We generate these videos with randomized camera position, contact points, force angles, and magnitudes. We use a mixed dataset with the goal of teaching the model that a point force can result in both simple linear motion, and complex oscillatory dynamics, depending on what type of object the force is applied to. In both scenarios, the force magnitude 0 corresponds to a very gentle poke, and the force magnitude 1 corresponds to a much stronger poke.

For both datasets, we project forces from 3D space onto the 2D pixel plane using the camera's parameters. This transformation maps force vectors and object positions from the physical world coordinate system to screen coordinates, allowing us to model forces within the image frame. We generate detailed text prompts using the GPT-4o API, creating a unique descriptions for each HRDI background and ground texture, plus a single shared prompt for all PhysDreamer carnation videos.

## 3.2 Local and Global Force Prompts

As the wind force is applied globally, and the point force is applied locally, we propose two different force encoding strategies.

**Encoding strategy, global force:** The wind force control signal is parameterized by a force $F \in [0, 1]$ and an angle $\theta \in [0, 360)$. The goal is to develop a tensor representation for the physics prompt $\pi$, which we denote by $\tilde{\pi} \in \mathbb{R}^{f \times c \times h \times w}$. Here, $f = 49$ is the number of frames, $c = 3$ is the number of color channels, and $h = 480$ and $w = 720$ are the height and width of the generated video. We define the first channel of $\tilde{\pi}$ to be $-1 + 2 \cdot F \in [-1, 1]$, the second channel to be $\cos \theta$, and the third angle to be $\sin \theta$. This defines a smooth map $[0, 1] \times [0, 360) \to \mathbb{R}^{f \times c \times h \times w}$ which encodes the angle and magnitude of the wind force field.

**Encoding strategy, local force:** The point force control signal $\pi$ specifies a localized force, so it is parameterized by the pixel coordinates $(x, y) \in \{0, \ldots, w - 1\} \times \{0, \ldots, h - 1\}$ in addition to the force magnitude $F \in [0, 1]$ and angle $\theta \in [0, 360)$. At a high level, we define the tensor

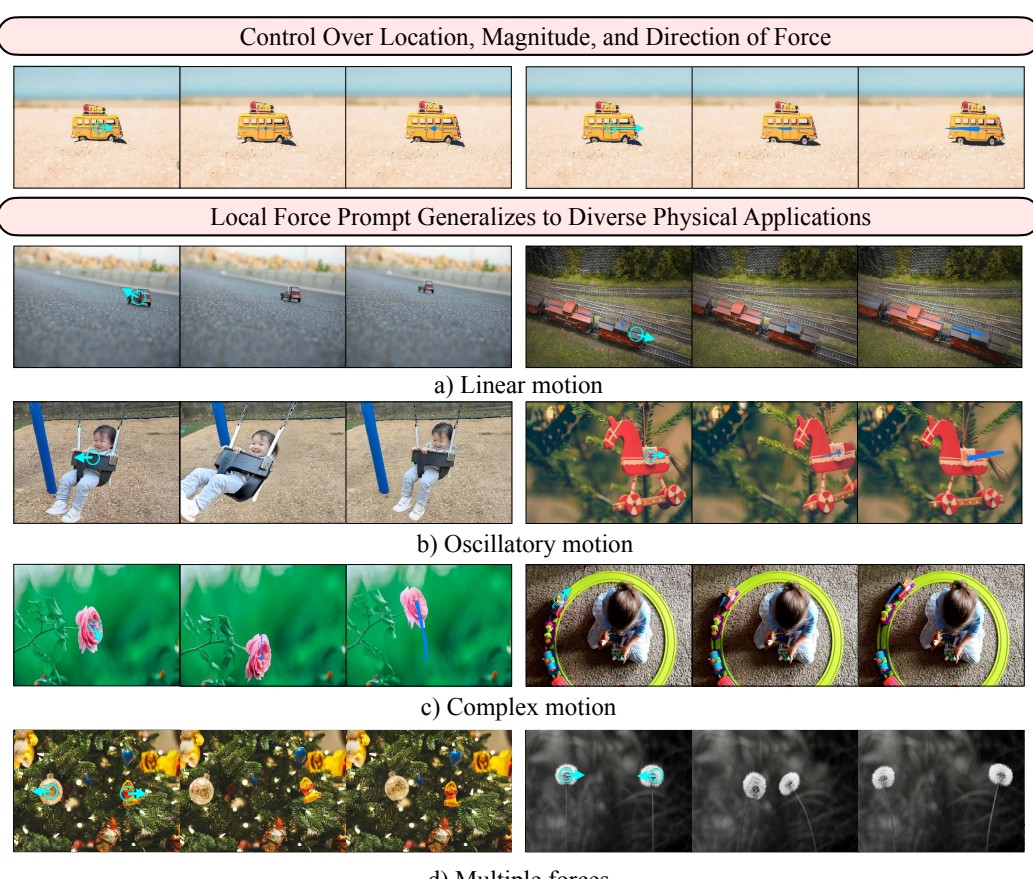

Figure 3: **Qualitative results for the Local Force (Poke) model**. *Top section:* For local forces, the control signal can specify both the location, magnitude, and direction of the force. *Bottom section:* despite the limited training data, the model generalizes to different types of motion. We add blue lines to visualize a time-lapse of some objects' movements.

representation $\tilde{\pi} \in \mathbb{R}^{f \times c \times h \times w}$ for the control signal $\pi$ to be a sequence of frames where a Gaussian blob starts at the pixel location $(x, y)$, and then moves in the direction $\theta$ at a constant velocity, for a total distance affinely proportional to the force $F$. Full mathematical details in Appendix A.3. This defines a continuous map $\{0, \ldots, w - 1\} \times \{0, \ldots, h - 1\}] \times [0, 1] \times [0, 360) \to \mathbb{R}^{f \times c \times h \times w}$ which encodes the coordinates where the force is applied, as well as the point force magnitude and angle, into a tensor representation $\tilde{\pi}$. We present a visual example of this in Figure 2. In the case of the local force, we note that the displacement of the Gaussian blob is nonzero when the force is $F = 0$, as our training dataset convention is that $F = 0$ indicates a small force.

We note that force values across the ball rolling and plant poking training videos are not calibrated to any absolute physical scale. Instead, they follow intuitive relative physics where smaller force values (approaching $F = 0$) correspond to gentle pokes resulting in minimal initial displacement, while larger force values produce stronger pokes with correspondingly greater initial displacement. We also wish to highlight that our force prompting models are fundamentally different from video generative models with trajectory-based control such as (Zhang et al., 2024b; Geng et al., 2024). This is because the gaussian blob which serves as the force indicator for the point force model is generally far away from the pixels that it affects, as demonstrated in the complex oscillatory motion of the swaying flower in Figure 2. Similarly, the wind force control signal under-specifies which points must move to which locations, as that control signal is global and causal.

| Point Force Model | Linear Motion | | | Oscillatory Motion | | | Complex Motion | | |
|---|---|---|---|---|---|---|---|---|---|
| | *Force Adh.* | *Real. Physics* | *Visual Qual.* | *Force Adh.* | *Real. Physics* | *Visual Qual.* | *Force Adh.* | *Real. Physics* | *Visual Qual.* |
| Text-only, zero-shot | 72% | 50% | 48% | 67% | 48% | 52% | 73% | 48% | 49% |
| Text-only, fine-tuned | 79% | 53% | 52% | 62% | 52% | 58% | 74% | 55% | 54% |
| Motion Prompting | 91% | 93% | 100% | 89% | 76% | 99% | 86% | 76% | 98% |

| Global Force Model | Tethered Motion | | | Aerodynamic Motion | | | Fluid Dynamics | | |
|---|---|---|---|---|---|---|---|---|---|
| | *Force Adh.* | *Real. Physics* | *Visual Qual.* | *Force Adh.* | *Real. Physics* | *Visual Qual.* | *Force Adh.* | *Real. Physics* | *Visual Qual.* |
| Text-only, zero-shot | 91% | 50% | 54% | 97% | 48% | 47% | 84% | 53% | 47% |
| Text-only, fine-tuned | 62% | 48% | 47% | 57% | 70% | 50% | 71% | 58% | 49% |
| Motion Prompting | 93% | 82% | 100% | 90% | 75% | 100% | 90% | 80% | 95% |

Table 1: **Comparison to baselines.** *Top:* Local point force model. *Bottom:* Global wind force model. We present % win rates of our method against baselines in 2AFC human study results (i.e. values above 50% indicate a preference for Force Prompting) for force adherence, realistic physics, and visual quality. We find that none of the other methods provide consistent adherence to the input force.

## 3.3 Architecture and Training

We build the force prompting models on top of CogVideoX-5B-I2V (Yang et al., 2024), a video generative model which accepts text and initial frame as conditional inputs. This model generates 49-frame videos at 8-fps. In order to integrate force prompt conditioning, we add a ControlNet (Zhang et al., 2023) which inputs a physics control prompt $\pi$, processing it through downscaling, encoding, and temporal compression before combining with hidden states via a zero convolution. The ControlNet clones the first six transformer layers and fine-tunes them while keeping the base model's transformer layers frozen. We base our implementation on (Karachev and Xu, 2025) with modifications to adhere more closely to the original ControlNet design. We train the models on a four 80 GB A100 GPU cluster for 5000 training steps, which takes approximately one day. Training uses an instantaneous batch size per device of $1$, with two gradient accumulation steps, for an effective batch size of $8$. Full hyperparameter details are listed in Appendix A.1.

## 4 Quantitative and Qualitative Results

We propose a benchmark dataset for both force prompting models using images that we curate from Pexels. We conduct a 2AFC human study ($N = 10$) using Prolific comparing our force prompting model against three baselines on these benchmark datasets.

**Baseline models:** The first baseline is *text-only, zero-shot*, which uses the original CogVideoX model and describes the intended force with a string and appending it to the end of the original text prompt. Two example prompt string suffixes are "the apple is moved very forcefully, upwards and to the left", and "the wind is medium strength, blowing right". The second baseline is *text-only, fine-tuned*, which has the same ControlNet architecture as our force prompting model, but with zero-tensor control signals, as well as force suffixes added to the end of the text prompts during training. Our third baseline is *Motion Prompting* (Geng et al., 2024), built on Lumiere (Bar-Tal et al., 2024) (run by the authors). It is the only track-conditioned model that accepts temporally sparse tracks as conditioning signal. We simulate force prompt's impulse by tracing push paths from target objects for the first 3 frames. While the model is meant to accept temporally sparse trajectories, 3 frames of trajectory is out of domain for the intended use case of Motion Prompting.

**Human study for local force benchmark:** We create a benchmark by curating 63 images from Pexels demonstrating three categories of physical interactions: 1) *linear* movement patterns (toy car, toy train on straight track, hot air balloon); 2) *oscillatory* movement patterns (windmill, pendulums, ornament, and swing); and 3) *complex* movement patterns (toy train on circular track, various plants including ivy, apple tree, and flowers). Table 1 presents human evaluation results for point forces, showing that despite training only on ball rolling (linear) and plant poking (complex) scenarios, our force prompting model demonstrates strong generalization across all motion categories. We note that this model successfully handles multiple forces "zero-shot" during inference, despite only being trained to handle a single force, as seen in Figure 3 and detailed in Appendix B.1.

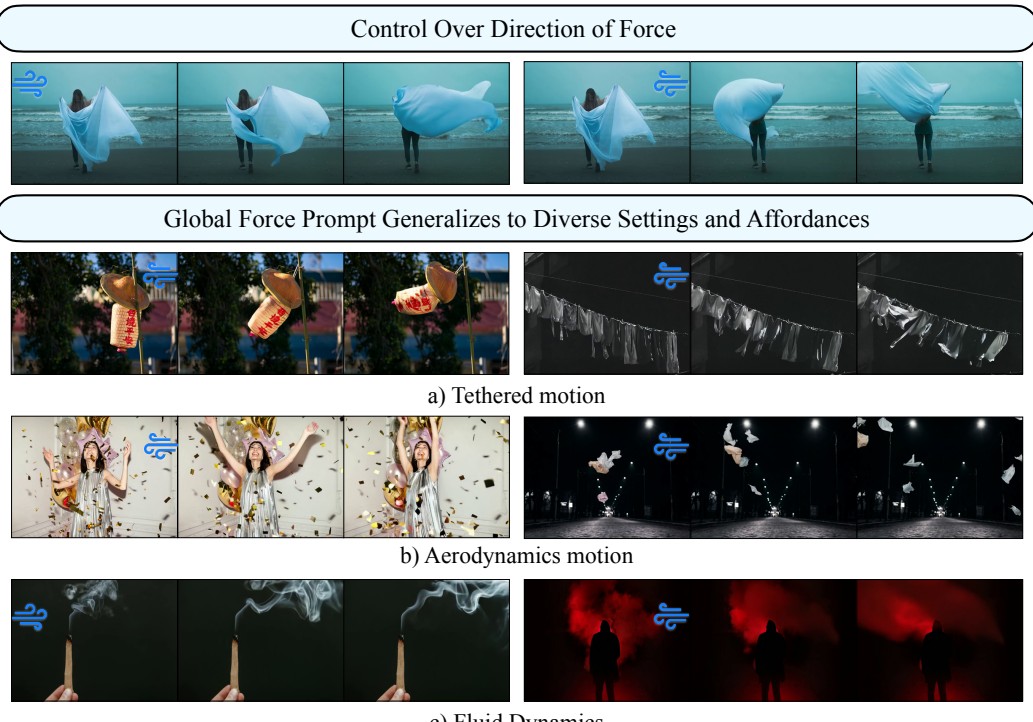

Figure 4: **Qualitative results for the Global Force (Wind) model**. *Top:* from the same starting image, different directions for the force result in different videos. *Bottom:* while the model was only trained on flags, it can generalize to many different settings producing different types of motion.

**Human study for global force benchmark:** We create a benchmark by curating 41 images from Pexels which demonstrate three different types of physical properties. The first is *tethered motion* (hair, cloth, clothing on person, paper lantern attached to hook). The second is *aerodynamic motion* (bubbles, falling leaves, inflatable tube in pool, floating litter, confetti). And the third is *fluid dynamics* (fog, smoke, snow, steam). In Table 1 we present human evaluation results for the global wind force model. Note that the base CogVideoX model is good at generating videos for all three motion categories (tethered, aerodynamic, fluid dynamic). However, our training data only has tethered motion (flags waving on a flagpole). We observe that the global wind control model trained only on labeled videos with tethered motion results in a model with generalized control over aerodynamic motion and fluid motion as well. We visualize some of these generalization patterns in Figure 4.

**Human study comparing to PhysDreamer:** The point force model, trained on data from a single carnation, demonstrates remarkable generalization to other plants, as we illustrate in Figures 1 and 3. To evaluate this generalization quantitatively, we compare our approach against PhysDreamer (Zhang et al., 2024a), which employs 3D assets and an integrated physics simulator. Using their benchmark dataset of six plant species, our results in Table 2 show that the point force model successfully generalizes to various roses, tulips, and alocasia without specific training on these plants. While we do not claim to replace physics-based simulation approaches, our purely neural method offers exceptional generalizability and produces responses that align with "intuitive physics," effectively conveying plausible physical interactions to human evaluators. We also also conduct an extended qualitative comparison with 8 other physics simulation models; see Appendix C.2 for more details.

## 5 Ablation Studies

### 5.1 Ablation Study #1: Composition of Synthetic Dataset

*How do synthetic dataset design choices affect model generalization?* In this section we analyze the impact of dataset diversity on force modeling tasks. Sample results are illustrated in Figure 5, more in depth results are in Figure 7 (Appendix), and additional videos are on the project webpage.

Ablation Studies: Importance of Strategic Diversity in Synthetic Training Data

| Global wind force model ablation: training videos contain **only 1 flag** | Global wind force model ablation: training videos contain **only 1 background** | Local point force model ablation: training videos contain **only 1 ball, no distractors** |
|---|---|---|

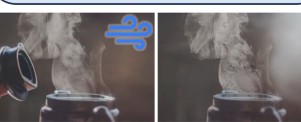 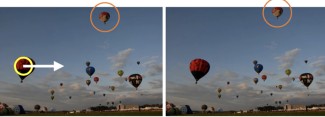

*Model overfits to waving flag, resulting in physically implausible arm motion* | *Degradation in background visual quality, control signal fails to generalize to steam control* | *Force prompt doesn't localize correctly, causing the wrong hot air balloon to move*

Figure 5: **Results from our ablation studies on synthetic dataset design choices.** *Left:* when the global wind force model is trained on a dataset with only one flag, it overfits, causing the woman's arm to wave unnaturally like fabric. *Middle:* when trained with a single background, the global force model has significantly degraded overall visual quality. *Right:* when trained without distractor objects, the point force model cannot properly localize motion, applying forces indiscriminately rather than to the intended target.

|  | Alocacia | Carnation | Rose (Orange) | Rose (Red) | Rose (White) | Tulip | **Mean** |
|---|---|---|---|---|---|---|---|
| Motion Realism | 40% | 50% | 50% | 60% | 40% | 50% | 48.33% |
| Visual Quality | 20% | 40% | 50% | 40% | 20% | 50% | 36.67% |
| Force Adherence | 60% | 70% | 50% | 50% | 50% | 70% | 58.33% |

Table 2: **Comparison to PhysDreamer.** Values represent the percentage of evaluators preferring the Force Prompting model over PhysDreamer, an approach that uses physics simulation during generation. Values above 50% indicate preference for our force prompting model. The results show that Force Prompting outperforms PhysDreamer on force adherence and achieves comparable performance on motion realism, while PhysDreamer maintains an advantage in visual quality.

**Point force training dataset ablation:** For the localized point force task, we conduct an ablation study by removing "distractor balls" from scenes, leaving only a single ball affected by the point force. Our results show that the presence distractor balls significantly improves force localization. Without them, the model exhibits undesirable behaviors: when poking one hot air balloon, all balloons move slightly; when poking a rose in a glass vase, both the rose and vase move together, failing to isolate the force application. Visuals are in Figure 7 as well as the project webpage.

**Global force training dataset ablation:** For the global wind force task, we evaluate two diversity factors: flag quantity and background variety. We find that training with a single background leads to models that follow force physics but frequently fail to differentiate between foreground and background, reducing visual quality. Similarly, when restricting scenes to contain only one flag instead of a variable number ($\text{Unif}\{1,\ldots,64\}$), the model successfully models cloth mechanics but fails to generalize to other materials. In these cases, smoke from campfires remains unaffected by wind, and confetti either doesn't respond or stays unnaturally suspended. We also observe that bubbles don't respond to wind, while human limbs incorrectly billow like cloth. These failures indicate that insufficient scene diversity causes the model to overfit to stationary backgrounds and limited material interactions. These findings are illustrated in Figure 7 as well as the project webpage. Additionally, we trained a unified model to learn both point force prompts and wind force prompts. We found that this results in more dynamic backgrounds, but has slightly less robust point force control. Additional details are in Appendix C.3.

## 5.2 Ablation Study #2: Text Prompt Specificity

*How does specificity of the text prompt affect model outputs?* In this ablation study, we investigate how material descriptions in text prompts affect model generalization through a $2 \times 2$ grid search ablation study. We train and test our wind model with and without wind-related keywords (wind/breeze/blow). Our results in Figure 8 and the project webpage show that omitting these keywords during training significantly increases failure cases in our benchmark dataset—fog remains static, lanterns collapse unexpectedly, and steam appears without cause. In contrast, models trained with wind-specific terminology demonstrate superior generalization to diverse wind scenarios. Interestingly, the presence of

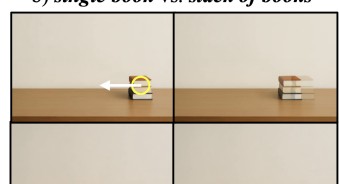

Figure 6: **Mass understanding:** We find that the model has some degree of understanding of mass, in that the same force applied to two objects with different masses will result in different amounts of motion. We demonstrate this qualitatively in (a) and (b) and quantitatively in (c), showing that this result is consistent across a range of force magnitudes. See additional examples in the project webpage.

these keywords during inference has less impact than during training, though using wind terminology generally produces more robust results.

# 6 Mass Understanding

This section examines the model's capacity for *mass understanding*, which we define to be the ability to recognize that objects with different apparent masses should respond distinctively to the same applied force. A model with robust mass understanding would demonstrate physically intuitive behaviors: a book sliding further than a stack of books when pushed with equal force, or a wooden ornament swinging more freely than its identical metal counterpart under the same impulse. We focus our quantitative analysis on the ball-rolling scenario, as it allows for objective measurement using automatic object detection. Then, we focus our qualitative analysis on other scenarios which present greater challenges for obtaining reliable metrics at scale, such as the swinging ornament.

**Force-Mass Relationship Quantitative Study:** To quantitatively assess the model's mass understanding, we design an experiment to measure whether soccer balls roll farther than bowling balls when subjected to identical forces. We generate initial condition images across four ground surfaces (dirt, grass, stone, and wood), with three color variations each for both bowling balls and soccer balls. Additional experiment details are in Appendix A.2. Results presented in Figure 6 confirm two key physical principles: the distance traveled increases linearly with applied force for both ball types, and soccer balls consistently travel farther than bowling balls across all force magnitudes, demonstrating the model's intuitive understanding of mass-dependent physics in this scenario.

**Force-Mass Relationship Qualitative Study:** We evaluate mass understanding across four benchmark tasks featuring geometrically identical objects with different implied masses. Our test scenarios includes ornaments (wooden versus cast iron), laundry baskets (empty versus filled with clothes), book stacks (one, two, or three books), and cube stacks (single versus double cube). To ensure experimental control, we utilize the GPT-Image-1 API to generate initial frames with variations where only the implied mass differs between conditions. Figure 6 presents some of these results, with demonstrating that lighter objects consistently travel farther when subjected to identical forces. This pattern remains robust across four random seeds. This behavior suggests an emergent understanding of mass-dependent physics in our force-prompted model. Other results are in the project webpage. Additionally, we find that the mass understanding behavior persists in the zero-shot multiple objects setting. We include these experimental details in Appendix B.2.

# 7 Conclusion

We introduce Force Prompting, enabling users to interact with generative video models through physically meaningful controls including localized point forces and global wind effects. Our approach demonstrates that video generation models can successfully learn to respond to force-based conditioning from limited synthetic training data, generalizing remarkably well to diverse objects, materials, and scenarios without requiring physics simulators at inference time. These results suggest a promising direction for developing intuitive world models that respond to natural physical interactions, with potential applications in both creative content generation and embodied AI planning.

## Acknowledgments

We would like to thank Bill Freeman, Miki Rubinstein, Junyi Zhang, Junhwa Hur, Noah Fischer, Saining Xie, Calvin Luo, Shijie Wang, Koven Yu, Tian Yun, and Zilai Zeng for useful discussions. We would also like to thank the anonymous NeurIPS reviewers. This project was partially supported by Samsung. Our research was conducted using computational resources at the Center for Computation and Visualization at Brown University.

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

# A  Implementation details

## A.1  Training hyperparameters

We train for 5000 steps with an effective batch size of 8 (using gradient accumulation over two steps) and saved model checkpoints every five hundred steps. We used `bf16` mixed precision with `tf32` support and initialized from the `THUDM/CogVideoX-5b-I2V` pretrained weights. ControlNet (Zhang et al., 2023) was initialized from the first six transformer layers, for all three input channels, using a downscaling factor of eight, and a unit weight coefficient. We use the AdamW optimizer (Loshchilov and Hutter, 2019) (initial learning rate $1 \times 10^{-5}$, $\beta_1 = 0.9$, $\beta_2 = 0.95$, maximum gradient norm 1.0) under a cosine-with-restarts learning-rate schedule (one cycle, 250 warm-up steps). We set our random seed to 42. Training videos contained up to 49 frames at 8 fps.

There are a total of 42 transformer blocks that we may use to initialize our ControlNet. We chose the first six due to memory constraints. With more compute resources, one could potentially achieve higher quality control. We expose the number of transformer blocks to use in the ControlNet as a command line argument for convenience in our training code release.

## A.2  Additional details: mass understanding quantitative study

Each ball is subjected to forces of magnitude $0.125 \cdot n$, where $n \in \{1, \ldots, 8\}$, with 10 videos generated per force value using different random seeds. To ensure experimental control, we utilize the GPT-Image-1 API to generate first frames and their variations, maintaining consistent initial ball positions and shapes across conditions. We automatically detect ball position using a Faster R-CNN model with a ResNet-50-FPN backbone (Ren et al., 2015), tracking the sports ball class (ID 37). We compute the distance traveled as the Euclidean distance in pixel space between detected bounding box centers in the first and last frames.

## A.3  Additional details: encoding strategy, point force

For the first frame, we set the pixel values equal to $0$ everywhere except for a Gaussian blob of radius 20 centered at $(x, y)$, which gets a value of $1$; and at the final frame of the control signal, the blob will have moved a distance of $(\frac{1}{8} + \frac{3}{8}F) \cdot w$ pixels. This ensures that when the force is minimized at $F = 0$, the total displacement is $w/8$, and when the force is maximized at $F = 1$, the total displacement is $w/2$.

## A.4  Additional details: comparison with PhysDreamer

We took six flower demo videos from the PhysDreamer teaser videos (Alocacia, Carnation, Orange Rose, Red Rose, White Rose, Tulip) and took a screenshot of a still frame from the video that did not have any force annotation on it. We passed these six frames, as well as the six equivalent force prompts (i.e. the same force vector from the original demo) into the Force Prompting model. We presented these videos side by side in the human study, which we served using Qualtrics. Both videos for a given scene were shown to the human annotator simultaneously, with left/right randomization.

# B  Additional details: zero-shot multiple forces

## B.1  Multiple forces for multiple objects, benchmark

Our experiments confirm that the model successfully handles multiple simultaneous forces without requiring retraining. This capability emerges zero-shot by simply adding multiple Gaussian blobs to the control signal videos—one for each applied force. We include these videos in the project webpage. The 6 scenes we tested are:

- An image with 2 Christmas tree ornaments
- An image with 2 toy cars
- An image with 2 vases, each with a flower in it
- An image with 2 roses leaning diagonally, growing out of the ground

- An image with 2 dandelions growing in a field
- An image with 2 apples on separate branches of a tree

We generated these images using the GPT-Image-1 API. To test each image, we identified two directions where it would be reasonable to poke each object; for example, the dandelions can each be poked to the left and to the right. Then for each image, we used 4 different multiple-force force prompts, representing the $2 \times 2$ different ways of combining the two different directions; for example, we can poke both dandelions to the left, poke both to the right, poke one to the right and the other to the left, then poke one to the left and the other to the right. We computed one video for each, using the same seed for all of them, with no cherry-picking. We found that $5/6$ of the videos in this multi-poke benchmark had perfect force adherence, and one of them had nearly perfect force adherence. The only failure case was in the two apples scene; when the left apple is poked to the right and the right apple is poked to the left, they move as if they were both poked to the right.

### B.2 Multiple forces for multiple objects, mass understanding

We used the GPT-Image-1 API to construct images where two objects of different apparent masses are present. The first image contains an empty laundry basket on the left side of the frame, and a full laundry basket on the right side of the frame. The second image contains a book on the left side of the frame, and a stack of books on the right side of the frame. We passed both sets of images into the multi-poke Force Prompting inference pipeline and instructed the model to poke both sets of objects towards the middle of the frame with the same force magnitude. Across 8 different force magnitudes $\{0.125 * i, i \in \{1, \ldots, 8\}\}$ we found that the lighter object moved much further, with the heavier item barely moving at all. We include videos on our project page.

## C Additional qualitative results

### C.1 Failures and Limitations

Figure 9 illustrates and categorizes failure cases of Force Prompting. We observe model correlation issues—for example, in hair-blowing scenarios, faces sometimes reorient based on wind direction, likely reflecting patterns in training data where hair typically blows backward. Our method is fundamentally constrained by the underlying video prior's physical understanding; we focus on controlling existing physical capabilities rather than improving the model's physics comprehension. We defer to other works that specifically aim to enhance physical accuracy in generative models, while noting that our approach benefits from efficiently leveraging the scaling properties of the base model.

### C.2 Extended comparison with physics simulation models

To demonstrate the point force model's versatility, we curate a benchmark using first-frame images from prominent physics-in-the-loop papers: PhysDreamer (Zhang et al., 2024a), DreamPhysics Huang et al. (2024), MotionCraft (Aira et al., 2024), PhysGaussian (Xie et al., 2024), PhysGen (Liu et al., 2024b), PhysGen3D (Chen et al., 2025), Physics3D (Liu et al., 2024a), and PhysMotion (Tan et al., 2024). We apply our force prompting approach to these diverse scenarios, including poking plants (alocasia, ficus, bouquet of flowers), moving vehicles (boat in water, toy cars), and household objects (rocking horse). Our video results (see project webpage) illustrate that our purely neural method can handle the same visual scenarios almost as effectively as approaches requiring explicit physics simulation at inference time. These qualitative results are in line with our findings in Table 2.

### C.3 Training a unified model

We also train a unified model to learn both point force prompts and wind force prompts. We run inference on this joint model using our point force benchmark, as well as our wind force benchmark. Our findings:

- *More dynamic backgrounds:* in many of the videos, the background moves more dynamically in a natural way. For example, for the apple tree, the surrounding leaves move more naturally

Ablation Studies: Importance of Synthetic Training Data Diversity

Global wind force model ablation: training videos **contain only 1 flag**

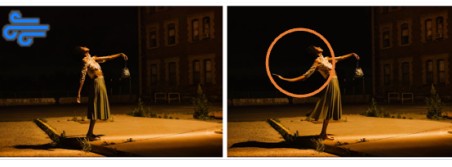
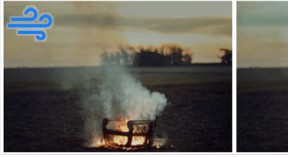
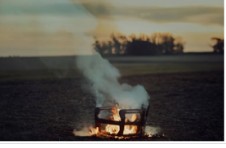

*Model overfits to waving flag, resulting in physically implausible arm motion*

*Control signal fails to generalize to smoke control*

Global wind force model ablation: training videos **contain only 1 background**

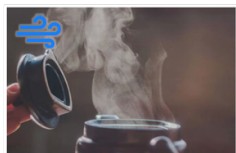
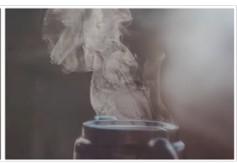
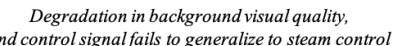
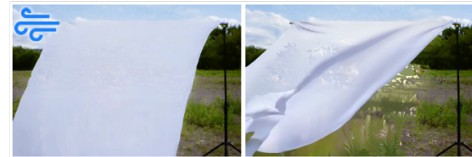

*Degradation in background visual quality, and control signal fails to generalize to steam control*

*Degradation in background visual quality*

Local point force model ablation: training videos **contain only 1 ball, no distractor balls**

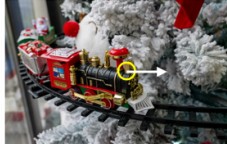
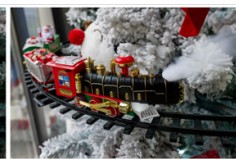
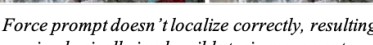
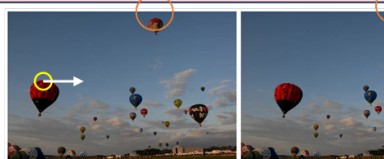

*Force prompt doesn't localize correctly, resulting in physically implausible train movement*

*Force prompt doesn't localize correctly, causing the wrong hot air balloon to move*

Figure 7: **Results from our ablation studies on synthetic dataset design choices.** *Top:* when the global wind force model is trained on a dataset with only one flag, it overfits, causing the woman's arm to wave unnaturally like fabric and failing to generalize to fluid dynamics scenarios such as smoke. *Middle:* when trained with a single background, the global force model fails to differentiate between foreground and background elements, significantly degrading overall visual quality. *Bottom:* when trained without distractor objects, the point force model cannot properly localize motion, applying forces indiscriminately rather than to the intended target.

> after the apple is poked; in the video where a kid is sitting in the middle of a toy train track, the kid is moving more naturally while the train is moving around the track; and in the video with falling leaves in the forest with a woman sitting on a chair in the background, the woman in the chair moves more while the leaves are being blown.

- *Slightly less robust point force control:* on some of the point force videos (e.g. the blueberry bush), the control signal is not respected.

We designed this experiment by sourcing 50% of the training data and control signals in each batch from the synthetic point force dataset, and the other 50% from the synthetic wind force dataset. We trained using the same architecture and number of training steps as the original model.

## C.4   Scaling the dataset size

We trained from scratch a wind force model which uses half as many synthetic flag waving videos. The only variable that we changed was the dataset size; everything else (including the number of training steps and learning rate scheduler) remained the same. For this model, we found some additional failure cases. One failure case: for the image where a woman holds a sheet on the beach,

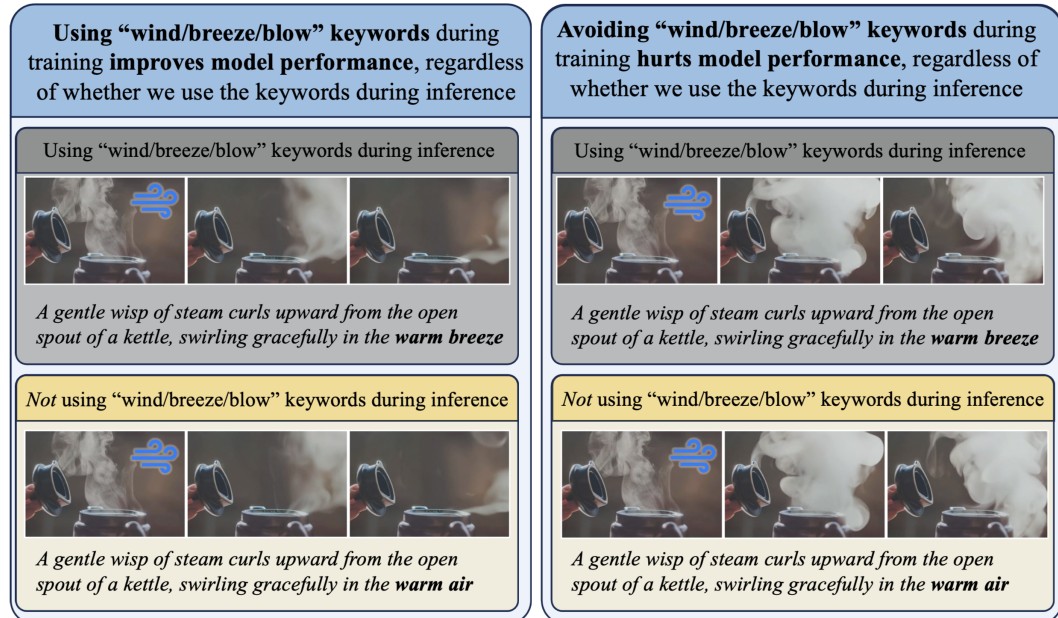

Ablation Studies: Importance of Using Force-Related Keywords

| **Using "wind/breeze/blow" keywords** during training **improves model performance**, regardless of whether we use the keywords during inference | **Avoiding "wind/breeze/blow" keywords** during training **hurts model performance**, regardless of whether we use the keywords during inference |

Figure 8: **Results from our ablation studies on text prompt specificity.** In this ablation study, we investigate how material descriptions in text prompts affect model generalization through a $2 \times 2$ grid search ablation study. We train and test our wind model with and without wind-related keywords (wind/breeze/blow). Our results demonstrate that omitting these keywords during training significantly increases failure cases in our benchmark dataset. For example, steam is conjured out of thin air instead of being blown correctly. In contrast, models trained with wind-specific terminology demonstrated superior generalization to diverse wind scenarios.

the model hallucinates a bedsheet in the background, indicating that the model has likely memorized the "waving flag" pattern from the training dataset and is injecting it into the output video. A second failure case: the confetti's response to the wind isn't as convincing. For example, some of the confetti will blow in the wind's direction but some will stay stationary. This indicates that the model hasn't generalized properly, perhaps because of less training data.

# D    Additional emergent phenomena

## D.1    Case study #1: Does the model enforce physical affordances?

The Force Prompting models demonstrate a surprising capability to respect object-specific movement constraints. For example, when a train on a circular track is poked forward, it follows the curved trajectory of the track rather than continuing in a straight line—a behavior similarly observed with windmills respecting their rotational axis. We also note interesting emergent behaviors with multi-part objects: poking the lead car of a toy train forward sometimes pulls the entire train along, while other times only the first car moves; conversely, backward forces consistently push the entire train as a unit.

## D.2    Case study #2: Does the model understand atomicity of objects?

We evaluate the local force model's sensitivity to the specific pixel chosen as the application point for localized forces. Results demonstrate consistent object movement regardless of which part of the object receives the force. For example, whether poking a train's engine, middle car, or caboose, the entire object responds appropriately to the applied force. This suggests the model has developed a holistic understanding of object wholeness rather than simply responding to pixel-level manipulations.

Limitations of Force Prompting Method

*Failure Case #1:* The Physics is Out-of-Domain For the Base Model

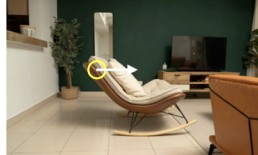
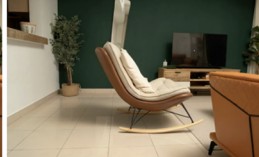
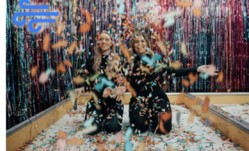
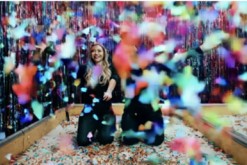

*The egg rolls in the prompted direction, but the base video model has difficulty rolling non-spherical objects, so the egg appears to float*

*The kite is blown in the prompted direction, but the base video model has difficulty generating a physically plausible video of a kite dragging a person*

*Failure Case #2:* The Base Video Model's Prior Competes with the Force Prompt

*The rocking chair moves in the prompted direction, but the base video model has trouble distinguishing between foreground and background objects*

*The confetti moves in the prompted direction, but the base video model forces the scene to conjure extra confetti*

Figure 9: **Analysis of failure cases.** We illustrate and categorize failure cases of Force Prompting. The *top row* shows scenarios where the generated physical motion is out-of-domain for the base CogVideoX model, leading to partial adherence to the force prompt. The *bottom row* depicts failures in visual fidelity or physical realism when the video prior conflicts with the force prompt's intent. More examples are available on our project webpage.

### D.3 Case study #3: Does the model preserve cinematic effects of the original image?

The model demonstrates an ability to maintain the original image's stylistic and cinematic properties throughout generated sequences. For example, when animating a toy car from an image with a depth-of-field effect, the model preserves the background blur as the car moves, ensuring visual continuity with the source image's aesthetic. This suggests the model not only understands physical motion but also respects the artistic intent and visual language of the input image. See the project page for videos.

## E Impact Statement

This work may be used to enhance the physical plausibility and controllability of video generation models. Applications include video content creation with fine-grained output control for physics-based forces and second-order solutions that leverage enhanced intuitive physics such as motion planning and world simulation. We urge the community to think critically about the potential risks of our work, specifically in the modeling of physical phenomena. Our work is not a substitute for precise physical simulation, rather we focus on what we have described as "intuitive physics", i.e. motion that is visually plausible to humans. Indeed, there are many *unintuitive* physical phenomena in the world where precise and specialist-level simulation is required. We emphasize that our work is unsuitable for use cases requiring high fidelity and precise simulation, including but not limited to materials science, architecture, mechanical engineering, and civil engineering.

Example 1. Please watch both videos. Which video more accurately shows the effect of wind **blowing to the right**?

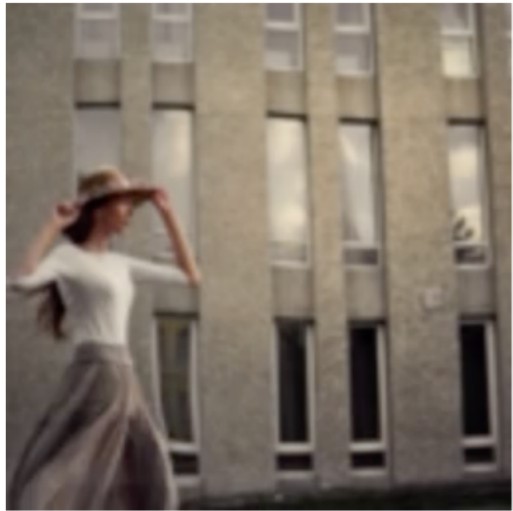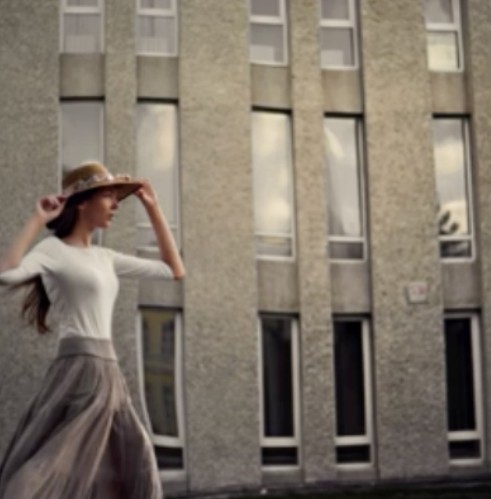

**Video 2** more accurately shows wind blowing to the right since the dress is blown to the right, while in Video 1 the dress is not blown at all.

Figure 10: **A demonstration question from one of our surveys.** Participants are shown an example question with a response along with the reasoning for that response.

## F  Survey Details and Instructions

We sourced participants from Prolific, compensating responders $12/hr. Our surveys specify the number of questions and an expected time limit. For example, we present the following to participants at the start of the survey:

> *Thank you for taking this survey!* ***It should take less than 25 minutes.***
>
> *There are 208 questions total. You should aim to spend around 7-8 seconds per question. You will be shown two videos, and you must choose which video more accurately* ***shows the effect of the wind blowing in the direction indicated by the question. Please read each question carefully.***
>
> ***There are hidden vigilance questions, so please make sure you answer to the best of your ability.*** *We will be rejecting extremely poor quality responses.*
>
> *At the end of the survey, there is a place to put your Prolific ID so we can confirm you've taken the survey.* ***Please respond only once to this survey*** *(you may have done a similar survey in the past day, that is fine) and thank you for your time!*
>
> *Please do not spend more than 25 minutes on this survey! We don't want to waste your time :)*

We then present participants with example questions and what we consider an appropriate response along with our reasoning, a screen shot of an example can be found in Figure 10. We then present participants with questions following the example for them to answer. They may select their preference from two videos by selecting the radio button underneath their selection, which is depicted in Figure 11.

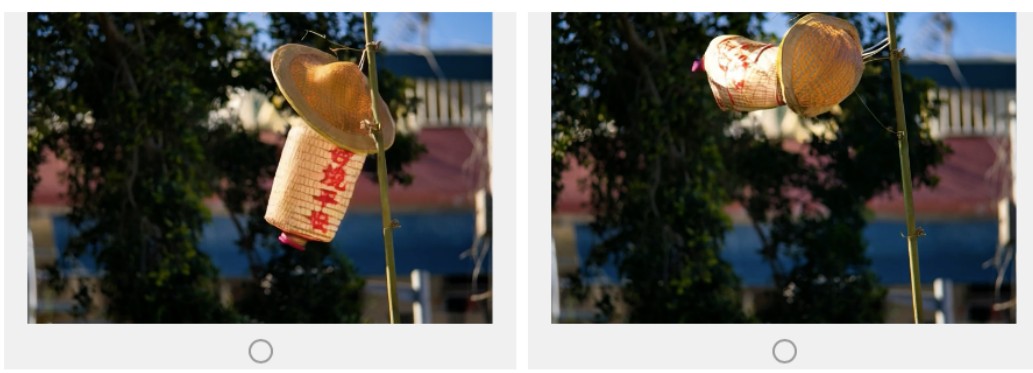

Figure 11: **A question from one of our surveys.** Participants are shown two videos side to side, with radio buttons beneath that they may use to make a selection of which better adheres to the question. The videos play automatically and simultaneously.

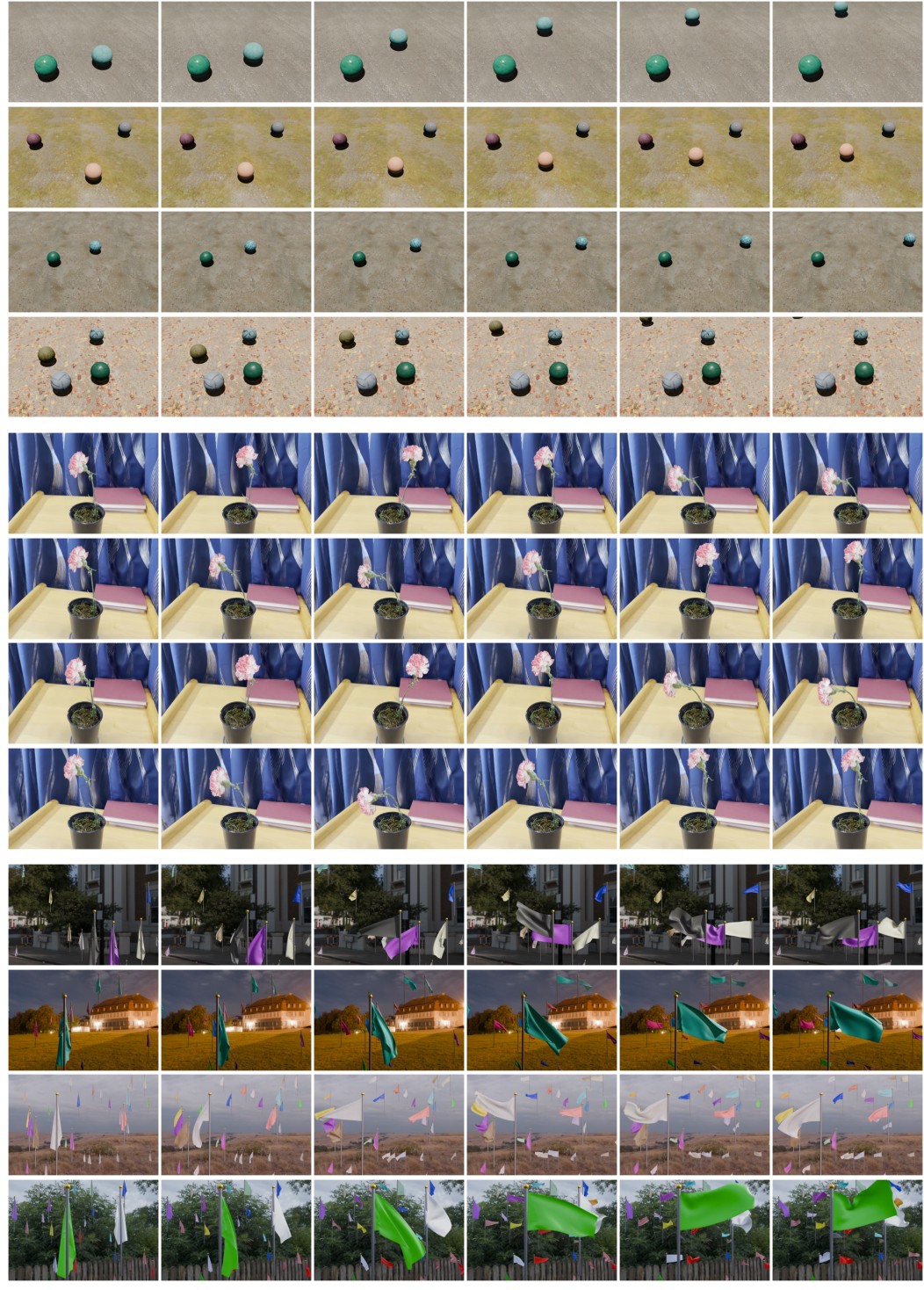

Figure 12: **Samples from our synthetic training datasets.** Top (ball) and middle (flower) are timelapses from our point force training dataset; bottom (flag) are timelapses from the global force training dataset. Our key finding is that video generation models can generalize well when adapted to follow physical force conditioning from videos synthesized by Blender, even with limited demonstrations on few ojects.

