# OpenReview forum: "Force Prompting: Video Generation Models Can Learn And Generalize Physics-based Control Signals"
_NeurIPS.cc/2025/Conference — NeurIPS 2025 poster_

### Official Review · Reviewer_dEDd · 2025-07-01

**Clarity:** 4
**Significance:** 4
**Originality:** 3
**Rating:** 5
**Confidence:** 5

**Summary:**

Force Prompting is a video generation framework that conditions a pretrained video generation model (CogVideoX) on explicit physical force signals (both global and localized forces) without requiring a physics engine at inference time. With ControlNet style finetuning, the CogVideoX-based video model can take in forces as conditions. The finetuning datasets are synthetically generated, using Blender scenes as well as an existing physics-based method (PhysDreamer). Through extensive human studies and automatic evaluations, the paper demonstrates that Force Prompting not only faithfully adheres to the specified forces but also generalizes beyond its limited synthetic training set to diverse real-world images, materials, and motions.

**Questions:**

How does performance improve (or plateau) if you scale up (or down) the synthetic dataset by 2×?
How confident are you about the mass understanding behavior? I think it cannot be conclusively determined until the method is extended to handle multiple objects, and then we can see how the model behaves when two objects of different mass are present in the same video.

**Ethical Concerns:**

["NO or VERY MINOR ethics concerns only"]

**Final Justification:**

I am happy with the response provided by the authors, and I will maintain the original rating to accept.

**Limitations:**

Yes

**Quality:**

4

**Strengths And Weaknesses:**

Strengths:
- Strong sim2real generalization despite limited training data.
- The methodology is sound, integrating a state-of-the-art video model with a lightweight ControlNet for physics conditioning.
- Experiments span multiple benchmarks, include ablations on data design and prompt specificity.
- Overall significance in demonstrating that one can achieve comparable visual effect to prior physics-engine-based works with a simple synthetic dataset and moderate finetuning costs (1 day of training with four A100 GPUs)

Weakness:
- Due to the nature of the method (based on video generation without modeling explicitly in 3D space), physical control (e.g. direction of forces) is not guaranteed to be adhered by the generated content by design.
- The method is limited by the capacity of the video generation model, which might lead to suboptimal performance on out-of-domain content.
- The length of the generated video is capped by the base model (CogVideoX).

---

> ### Author Rebuttal · Authors · 2025-07-31
>
> Thank you for the thoughtful feedback. Below we provide detailed answers to your questions.
>
> ---
>
> ## Q1: can the method be extended to handle multiple objects?
>
> **Additional experiments confirm that the model successfully handles multiple simultaneous forces without requiring retraining.** This capability emerges zero-shot by simply adding multiple Gaussian blobs to the control signal videos—one for each applied force. While submission constraints prevent us from updating the supplemental materials or sharing new videos at this time, we can provide detailed descriptions of these results and will make the videos available in the next version of our project website.
>
> The 6 scenes we tested are:
>
> * An image with 2 christmas tree ornaments
> * An image with 2 toy cars
> * An image with 2 vases, each with a flower in it
> * An image with 2 roses leaning diagonally, growing out of the ground
> * An image with 2 dandelions growing in a field
> * An image with 2 apples on separate branches of a tree
>
> To test each image, we identified two directions where it would be reasonable to poke each object (e.g. the dandelions can each be poked to the left and to the right). Then for each image, we used 4 different “multi-poke” force prompts, representing the 2x2 different ways of combining the two different directions (e.g. poke both dandelions to the left, poke both to the right, poke one to the right and the other to the left, then poke one to the left and the other to the right). We computed one video for each, using the same seed for all of them, with no cherry-picking allowed. We then computed the “multi-poke force adherence” to be the fraction of the 8 pokes which were respected.
>
> **We found that 5 / 6 of the videos in this multi-poke benchmark had perfect force adherence, and one of them had nearly perfect force adherence.** The only failure case: in the 2 apples scene, when the left apple is poked to the right and the right apple is poked to the left, they move as if they were both poked to the right.
>
> We want to emphasize that this worked zero-shot: no additional training to the model was required. In fact, we were prepared to train an additional model using synthetic videos of multiple balls being poked simultaneously, but decided to check if the desired behavior would work zero-shot instead. We report the results in the table below.
>
> |               Scene description        | 2 ornaments | 2 toy cars | 2 flowers in vases | 2 roses in nature | 2 dandelions | 2 apples |
> | :-------------------- | :---------- | :--------- | :----------------- | :---------------- | :----------- | :------- |
> | **Multi-poke force adherence** | 100%        | 100%       | 100%               | 100%              | 100%         | 87.5%    |
>
> We’ll include the experiment details in the next draft of the paper, and we’ll include the videos in the next draft of the project page.
>
> ---
>
> ## Q2: does mass understanding behavior persist in the multiple objects setting?
>
> **We have conducted new experiments and concluded that the mass understanding behavior indeed persists in the zero-shot multiple objects setting.** In more detail: we have used the GPT-Image-1 API to construct images where two objects of different apparent masses are present. The first image contains an empty laundry basket on the left side of the frame, and a full laundry basket on the right side of the frame. The second image contains a book on the left side of the frame, and a stack of books on the right side of the frame. We passed both sets of images into the multi-poke Force Prompting inference pipeline and instructed the model to poke both sets of objects towards the middle of the frame with the same force magnitude. Across 8 different force magnitudes $0.125*i,$ for  $i=1,\dots,8$, we found that the lighter object moved much further, with the heavier item barely moving at all. We will include the results on the next draft of the project page, as well as on the website.
>
> ---
>
> ## Q3: how does performance change if we scale up or down the synthetic training dataset?
>
> **We have conducted additional ablation studies, and we found that model robustness gets worse when you scale down the synthetic training dataset.** In more detail: we trained from scratch another wind force model. This new model uses half as many synthetic flag waving videos as the dataset that we used in the paper. The only variable that we changed was the dataset size; everything else (including the number of training steps and learning rate scheduler) remained the same.
>
> For this 0.5x dataset model, we found some additional failure cases. *One failure case:* for the image where a woman holds a sheet on the beach, the model hallucinates a bedsheet in the background, indicating that the model has likely memorized the “waving flag” pattern from the training dataset and is injecting it into the output video. *A second failure case:* the confetti’s response to the wind isn’t as convincing, e.g. some of the confetti will blow in the wind’s direction but some will stay stationary. This indicates that the model hasn’t generalized properly, perhaps because of less training data. Unfortunately we’re not allowed to share videos during the rebuttal period, but we will include these ablation experiments in the next draft of the paper.
>
> ---
>
> ## Q4: method is limited by capacity of video gen model, similarly with length of its outputs
>
> This is a good point. Our perspective is that this is a feature, rather than a bug. Our results in Table 1 indicate that Force Prompting videos are as physically realistic as the videos from the base model. This implies that as video models continue to improve, training a Force Prompting ControlNet on top of them will result in more physics capabilities. We hope that our results will become even more useful when applied to the larger, more capable video models of the future.

---

> ### Author Response · Authors · 2025-08-03
>
> We wanted to follow up to make sure that your concerns are being properly addressed. Please let us know if there are additional questions that we can provide further information or experiments on. Thank you again for your time and effort reviewing our paper!

---

> > ### Comment · Reviewer_dEDd · 2025-08-04
> >
> > Thank you for the detailed response, which I see as satisfactory. I will maintain the rating to accept this paper.

---

### Official Review · Reviewer_Pchz · 2025-07-02

**Clarity:** 4
**Significance:** 3
**Originality:** 3
**Rating:** 4
**Confidence:** 4

**Summary:**

The paper proposes a controllable image-to-video generation model conditioned on input forces, which include a point-force control model and global wind force model. To train the model, the author collected synthetic datasets using blender and found a limited amount of synthetic data is enough for training. The trained model also demonstrate good generalization across motion types and settings. User study shows the effectiveness of proposed method.

**Questions:**

1. **Interactive motion**

I wonder could such controllable model applied to model the interaction between the objects which applied force on and rest objects in the scene. For example, in the ball in the scene, can the model handle the ball collision when applying force on one ball towards the other.


2. **Unified model**

I am curious if possible to train a unified model that could handle both forces. The unified model is more helpful as current each model seems a little overfitting to certain scenarios. If trained in a unified way, will the model performance degrade and will the generalization ability disappear.

3. **Multi-objects**

In all existing demos shown in the paper, there is only one single object being affected by the force. I understand the author only collect a dataset with single objects, but the proposed control signal should have the ability to model multi objects at once (like simultaneously move multi balls), otherwise the apply setting is little weak, just like physgaussian, I think video generation models should have stronger capacity.

**Ethical Concerns:**

["NO or VERY MINOR ethics concerns only"]

**Final Justification:**

Thanks the author for the detailed rebuttal. I think the rebuttal experiments are convincing and solid. The limitation of the work is the motion is relatively simple without object-object interactions, but the model is still interesting and will benefit the community. Another issue is the lack of quantitative comparison, hope in the final version could see more comparison results to different method on more examples and rebuttal experiment settings.

**Limitations:**

yes

**Quality:**

3

**Strengths And Weaknesses:**

**Strengths**:

1. **Simple idea, good results**

The idea in the paper is simple but effective, and the visual results generated by the trained model are great.

2. **Synthetic dataset**

The model is fine-tuned using synthetic data, and the trained model could generalize to different real settings. It looks promising and provides good insight to the video generation community.

3. **Good experiments and ablation**

I appreciate the abundant experiments and ablation in the paper, and those summarized findings are interesting and helpful.

**Weakness:**

1. **Insufficient metrics**

All quantitative evaluation used in the paper are human user study, which is quite subjective. I would like to see some objective metrics. As the training using synthetic dataset, a straight way is to compare the generated result and simulated results (like moving ball) and evaluate the material influence, and mass understanding in a more general and rigorous way. Otherwise it could be hard to tell the results are occasion for certain examples or indeed some common patterns behind it.

2. **Missing qualitative comparison to different methods**

I didn't find the video results and qualitative comparison in B.2 and attached supp as claimed in the paper. The comparison to baseline cogvideox is necessary as it could show the improvement of introducing such explicit control signal.

3. **Limited setting**

From current results, it seems the model could only model one kind of force for each type of input, and only moving one object with simple motion (no interaction with the scene), the setting is quite constrained under my view.

---

> ### Author Rebuttal · Authors · 2025-07-31
>
> Thank you for the thoughtful feedback. Below we provide detailed answers to your questions.
>
> ---
>
> ## Q1: can the model be extended to handle multiple forces at once?
>
> **Additional experiments confirm that the model successfully handles multiple simultaneous forces without requiring retraining.** This capability emerges zero-shot by simply adding multiple Gaussian blobs to the control signal videos—one for each applied force. While submission constraints prevent us from updating the supplemental materials or sharing new videos at this time, we can provide detailed descriptions of these results and will make the videos available in the next version of our project website.
>
> The 6 scenes we tested are:
>
> * An image with 2 christmas tree ornaments
> * An image with 2 toy cars
> * An image with 2 vases, each with a flower in it
> * An image with 2 roses leaning diagonally, growing out of the ground
> * An image with 2 dandelions growing in a field
> * An image with 2 apples on separate branches of a tree
>
> To test each image, we identified two directions where it would be reasonable to poke each object (e.g. the dandelions can each be poked to the left and to the right). Then for each image, we used 4 different “multi-poke” force prompts, representing the 2x2 different ways of combining the two different directions (e.g. poke both dandelions to the left, poke both to the right, poke one to the right and the other to the left, then poke one to the left and the other to the right). We computed one video for each, using the same seed for all of them, with no cherry-picking allowed. We then computed the “multi-poke force adherence” to be the fraction of the 8 pokes which were respected.
>
> **We found that 5 / 6 of the videos in this multi-poke benchmark had perfect force adherence, and one of them had nearly perfect force adherence.** The only failure case: in the 2 apples scene, when the left apple is poked to the right and the right apple is poked to the left, they move as if they were both poked to the right.
>
> We want to emphasize that this worked zero-shot: no additional training to the model was required. In fact, we were prepared to train an additional model using synthetic videos of multiple balls being poked simultaneously, but decided to check if the desired behavior would work zero-shot instead. We report the results in the table below.
>
> |               Scene description        | 2 ornaments | 2 toy cars | 2 flowers in vases | 2 roses in nature | 2 dandelions | 2 apples |
> | :-------------------- | :---------- | :--------- | :----------------- | :---------------- | :----------- | :------- |
> | **Multi-poke force adherence** | 100%        | 100%       | 100%               | 100%              | 100%         | 87.5%    |
>
> Additionally, **we have conducted new experiments and concluded that the mass understanding behavior indeed persists in the zero-shot multiple objects setting.** In more detail: we have used the GPT-Image-1 API to construct images where two objects of different apparent masses are present. The first image contains an empty laundry basket on the left side of the frame, and a full laundry basket on the right side of the frame. The second image contains a book on the left side of the frame, and a stack of books on the right side of the frame. We passed both sets of images into the multi-poke Force Prompting inference pipeline and instructed the model to poke both sets of objects towards the middle of the frame with the same force magnitude. Across 8 different force magnitudes $0.125*i,$ for  $i=1,\dots,8$, we found that the lighter object moved much further, with the heavier item barely moving at all.
>
> We’ll include the experiment details in the next draft of the paper, and we’ll include the videos in the next draft of the project page.
>
> ---
>
> ## Q2: can a unified model be trained?
>
> **Additional experiments confirm that we can train a unified model to learn both point force prompts and wind force prompts.** We have trained a unified model which loads both datasets at once and trains on them jointly. We ran inference on this joint model using our point force benchmark, as well as our wind force benchmark. Our findings:
>
> * *More dynamic backgrounds:* in many of the videos, the background moves more dynamically in a natural way. For example, in the apple tree, the surrounding leaves move more naturally after the apple is poked; in the video where a kid is sitting in the middle of a toy train track, the kid is moving more naturally while the train is moving around the track; in the video with falling leaves in the forest with a woman sitting on a chair in the background, the woman in the chair moves more while the leaves are being blown.
> * *Slightly less robust point force control:* on some of the point force videos (e.g. the blueberry bush), the control signal is not respected.
>
> In more detail: we designed this experiment by sourcing 50% of the training data and control signals in each batch from the synthetic point force dataset, and the other 50% from the synthetic wind force dataset. We trained using the same architecture and number of training steps as the original model. We will include this ablation study in the next draft of the paper.
>
> ---
>
> ## Q3: request for more objective metrics
>
> We included objective metrics (not human study metrics) to evaluate how general the mass understanding behavior is in Figure 5. That graph demonstrates that (light) soccer balls roll much further than (heavy) bowling balls, and that this phenomenon is robust across ball color, ground material, and force magnitude. We copy the numbers here as well:
>
> | Force magnitude  | 0.125  | 0.250  | 0.375  | 0.500  | 0.625  | 0.750   | 0.875   | 1.000 |
> | :-------------- | :----- | :----- | :----- | :----- | :----- | :------ | :------ | :------ |
> | Distance traveled (bowling ball) | 37.75  | 48.49  | 61.86  | 73.07  | 88.06  | 100.01  | 115.83  | 135.38  |
> | Distance traveled (soccer ball) | 62.83  | 76.21  | 91.66  | 107.72 | 129.55 | 157.94  | 178.22  | 200.92  |
>
> Distances in this table are measured in pixels.
>
> ---
>
> ## Q4: missing qualitative comparison as promised in B.2
>
> **Table 1** demonstrates that CogVideoX+Force Prompting indeed has better adherence to the force prompt than the base CogVideoX model. Specifically, for the point force model, Force Prompting is preferred over CogVideoX more than 67% of the time, and for the wind force model, Force Prompting is preferred more than 84% of the time. These metrics are in the “Text-only, zero-shot” row; we’ll make it more clear in the next version of the paper that these come from the CogVideoX baseline. Additionally, in the next version of the project webpage, we’ll include qualitative examples for the base CogVideoX model.
>
> Not including the comparison to prior works as promised in B.2 was an oversight on our part, we apologize. We’re unfortunately not allowed to update the supplemental, but we can describe the videos to you, and we will release them in the next draft of the website (per neurips policy, all of our promises will be made public for the community’s inspection). The 8 papers and scenes we considered are:
>
> * PhysDreamer (ECCV 2024), poking an alocacia plant
> * DreamPhysics (AAAI 2025), poking a houseplant
> * MotionCraft (NeurIPS 2024), poking a ship in the water
> * PhysGaussian (CVPR 2024), poking a bouquet of flowers
> * PhysGen (ECCV 2024), poking a toy car
> * PhysGen3D (CVPR 2025), poking a toy car made of felt
> * PhysMotion, poking a toy rocking horse
> * Physics3D, poking a bouquet of flowers
>
> For each of those scenes, we take a screenshot of the first frame from the respective paper’s demo, and pass it through the Force Prompting model. We obtain a resulting video comparable to corresponding video from each of those sites, but without the need for any explicit 3D assets or physics simulation at inference time.

---

> > ### Comment · Reviewer_Pchz · 2025-08-05
> >
> > Thanks the author for the rebuttal. I read the rebuttal, I would recommend adding more quantitative experiments (more than 1example per experiment also concrete evaluation besides user study) to judge the performance, like if it could accurate mimic the "physics simulation", as well as filling those missing qualitative comparison results as promised in later version. It will be nice to add some comparison with physical simulation for the Q3 to see how good it is, instead of a human judgement of some numbers. I also understand the policy, but the added settings and missing qualitative comparison is little difficult for me to to judge with pure text.

---

> ### Author Response · Authors · 2025-08-03
>
> We wanted to follow up to make sure that your concerns are being properly addressed. Please let us know if there are additional questions that we can provide further information or experiments on. Thank you again for your time and effort reviewing our paper!

---

> ### Author Response · Authors · 2025-08-08
>
> Thanks for the suggestion. As you recommended, we have run another quantitative experiment to analyze how close the model mimics true physics simulation. **This experiment shows that the error between Force Prompting and a physics simulator is less than 9% for the bowling ball, and less than 12% for the soccer ball, across all force magnitudes.** The results are in the table:
>
> | Force magnitude  | 0.125  | 0.250  | 0.375  | 0.500  | 0.625  | 0.750   | 0.875   | 1.000 |
> | :-------------- | :----- | :----- | :----- | :----- | :----- | :------ | :------ | :------ |
> | Distance traveled (bowling ball, Force Prompting) | 37.75  | 48.49  | 61.86  | 73.07  | 88.06  | 100.01  | 115.83  | 135.38  |
> | **Distance traveled (bowling ball, simulator)** | 37.75 | 51.70 | 65.64 | 79.59 | 93.54 | 107.49 | 121.43 | 135.38 |
> | **Distance traveled % error (bowling ball)** | 0.00% | 6.20% | 5.76% | 8.19% | 5.86% | 6.96% | 4.61% | 0.00% |
> | Distance traveled (soccer ball, Force Prompting) | 62.83  | 76.21  | 91.66  | 107.72 | 129.55 | 157.94  | 178.22  | 200.92  |
> | **Distance traveled (soccer ball, simulator)** | 62.83 | 82.56 | 102.28 | 122.01 | 141.74 | 161.47 | 181.19 | 200.92 |
> | **Distance traveled % error (soccer ball)** | 0.00% | 7.69% | 10.39% | 11.71% | 8.60% | 2.18% | 1.64% | 0.00% |
>
>  (Bolded rows represent new computations that we ran.) Below we include the details of the experiment.
>
> ---
>
> ## Experiment details
>
> To model simulator behavior, we derived formulas for how far a sphere of mass M with radius R rolls when it is poked with a force of magnitude F for one second, then coasts for 5 seconds. The total distance for a bowling ball is $\frac{55F}{14M}$, and for a soccer ball is $\frac{33F}{10M}$. (Proof at the end.)
>
> In particular, a force of magnitude 0 results in no movement. In the Force Prompting model, the valid forces are in $[0,1]$, and our training data convention is that a force prompt with magnitude $0$ results in a small but nonzero amount of movement; accordingly, we consider affine variants of these formulas. In particular, we model the distance that a bowling ball travels as $d = \frac{55F}{14M}+d_0$, and the distance that a soccer ball travels as $d = \frac{33F}{10M}+d_0$. By plugging in minimum (F=0.125) and maximum (F=1.0) force values from our Force Prompting model, we solved for the ground truth simulator values for mass $M$ and offset distance $d_0$.
>
> The resulting formulas compute the distance a physics simulator would have the balls move:
> - **Bowling ball:** $d = \frac{55F}{14M}+d_0$ where $M=0.035,d_0=23.80$
> - **Soccer ball:** $d = \frac{33F}{10M}+d_0$ where $M=0.021,d_0=43.10$.
>
> These formulas yield the second and fifth rows in the table, respectively.
>
> ---
>
> ## Newtonian mechanics derivation
>
> We now provide the Newtonian mechanics derivation. We first consider the bowling ball. We model the bowling ball as a solid sphere of mass M and radius R, and we assume that a force F is applied to the ball uniformly for one second, and then no more force is applied and it coasts for 5s. (This is the setting considered in our training data.) We will show that the total distance traveled is $\frac{55F}{14M}$.
>
> ### Phase 1: acceleration for 1 second while rolling
>
> The only forces on the ball are the user-applied force $F$, and the friction force $f$. Newton’s second law implies that the net horizontal force is $F - f = Ma$, where $f$ is the backwards force from friction, and $a$ is the translational acceleration. The friction of the ball with the ground provides rotational motion. On one hand,  the torque is $\tau=fR$; on the other hand, the torque is $\tau=I\alpha$, where $I=\frac{2}{5}MR^2$ is the moment of inertia of a solid sphere, and $\alpha=a/R$ is the angular acceleration. Taken together, this implies that $fR=I\alpha$, which simplifes to $f=\frac{2}{5}Ma$ as the friction force. This implies that $F-\frac{2}{5}Ma=Ma$, so $a=\frac{5F}{7M}$. Finally, we can compute that the distance traveled after $t=1$ second is $d_1=\frac{1}{2}at^2 = \frac{5F}{14M}$, and the velocity at time $t=1$ is $v=at=\frac{5F}{7M}.$
>
> ### Phase 2: coasting for 5 seconds while rolling
>
> The distance traveled for $t=5$ seconds is $d_2=vt=\frac{25F}{7M}$.
>
> ### Sum of distance traveled in both phases
>
> Finally, we can compute that the total distance traveled for the bowling ball is $d=d_1+d_2 = \frac{55F}{14M}$.
>
> ## Adapting this to the case of the soccer ball
>
> The only difference is that we model the soccer ball as a hollow sphere, which has moment of inertia $I=\frac{2}{3}MR^2$. If we propagate this change through the previous derivation, we get a friction force $f=\frac{2}{3}Ma$, which results in acceleration $a=\frac{3F}{5M}$. The distance traveled in the first phase becomes $d_1=\frac{3F}{10M}$, the distance in the second phase becomes $d_2=\frac{3F}{5M}$, and so the total distance traveled is $d=d_1+d_2 =\frac{33F}{10M}.$ This completes the derivation.

---

### Official Review · Reviewer_ugp2 · 2025-07-03

**Clarity:** 3
**Significance:** 3
**Originality:** 3
**Rating:** 5
**Confidence:** 4

**Summary:**

The paper presents a study on learning video generation models from generalized physics-based control signals. Unlike previous force-controlled video generation methods that rely on integrated physical simulators, this work adopts a fully data-driven approach, enabling controllable interaction with images through force inputs. The authors introduced two types of force models: localized point forces and global wind force fields. To train their model, they constructed a dataset using diverse physical simulators from multiple sources. Extensive experiments demonstrated the model’s strong generalization capabilities in video generation.

**Questions:**

1. Line 145–146: What exactly does “HDRIs” refer to? Does it stand for High Dynamic Range Images? A brief explanation would be helpful for clarity.
2. In Table 1, why is PhysDreamer not included as a baseline for comparison?
3. In Table 2, the authors compare their method with PhysDreamer, but the explanation of the comparison is somewhat lacking. Since generalization is a key strength of this work, more evidence and clarification on how the comparisons were conducted would be valuable. Were the evaluation metrics and testing scenarios consistent?
4. To be honest, I had difficulty fully understanding the meaning of localized point forces and global force fields. Could the authors clarify why these terms were chosen? Are there more standard or physics-friendly terms that could be used to describe them?

**Ethical Concerns:**

["NO or VERY MINOR ethics concerns only"]

**Final Justification:**

After carefully reading the authors’ rebuttal and follow-up clarifications, I am now more confident in the contributions of this work. The authors have thoroughly addressed my major concerns, especially those related to dataset construction, baseline comparisons, and the rationale behind the choice of simulation tools.

I suggest the authors include in their revised manuscript the reasoning provided on why PhysDreamer and PhysGen were not ideal baselines for certain evaluations. I also understand that WonderPlay was released after the NeurIPS submission deadline, but it would be helpful to briefly acknowledge it in the revised version for completeness.

One area I found a bit unclear in the original submission was the choice of terminology—specifically, “localized point force” and “global force fields.” It would improve clarity if the paper could offer a more explicit definition and justification for these terms earlier on.

Despite these minor issues, I believe the novelty, feasibility, and potential impact of this paper clearly outweigh the concerns. Therefore, I raise my rating to 5: Accept.

**Limitations:**

1. The paper relies too heavily on existing paired force-video datasets and does not take advantage of larger force-enabled but unlabeled (unsupervised) datasets.
2. While the authors mention alternative approaches involving physical simulator integration, it’s unclear why comparisons with other existing methods such as PhysGen are not included. Could the authors elaborate on this choice?

**Paper Formatting Concerns:**

1. many of the referenced figures or supplementary materials in the paper do not hyperlink correctly when clicked, which affects the ease of review.
2. I noticed that the authors included the supplementary material at the end of the main paper. I’m not sure if this complies with the NeurIPS submission guidelines, but I raise this for the reviewers’ collective consideration.

**Quality:**

3

**Strengths And Weaknesses:**

Strengths

1. The paper proposes a fully data-driven approach that enables controllable interaction with visual content through force-based inputs.
2. A diverse dataset is constructed using multiple physical simulators, enhancing the robustness and variety of training data.
3. Extensive experiments convincingly demonstrate the model’s strong generalization ability in video generation tasks.

Weaknesses

1. The paper relies too heavily on existing paired force-video datasets and does not take advantage of larger force-enabled but unlabeled (unsupervised) datasets.
2. Line 145–146: What exactly does “HDRIs” refer to? Does it stand for High Dynamic Range Images? A brief explanation would be helpful for clarity.
3. In the construction of the local force dataset, the authors mention simulating a rolling ball in response to unseen point-wise forces using Blender. This raises a question: why was Blender chosen over other platforms like Unity or more advanced physics simulators?
4. A small portion of the dataset is collected using PhysDreamer. Given the limitations of existing simulators, will the proposed method face bottlenecks related to data diversity or label quality? How do the authors ensure high-quality, task-relevant force annotations for such specific downstream applications?
5. In Table 1, why is PhysDreamer not included as a baseline for comparison?
6. In Table 2, the authors compare their method with PhysDreamer, but the explanation of the comparison is somewhat lacking. Since generalization is a key strength of this work, more evidence and clarification on how the comparisons were conducted would be valuable. Were the evaluation metrics and testing scenarios consistent?
7. While the authors mention alternative approaches involving physical simulator integration, it’s unclear why comparisons with other existing methods such as PhysGen are not included. Could the authors elaborate on this choice?
8. Additionally, many of the referenced figures or supplementary materials in the paper do not hyperlink correctly when clicked, which affects the ease of review.

---

> ### Author Rebuttal · Authors · 2025-07-31
>
> Thank you for the thoughtful feedback. Below we provide detailed answers to your questions.
>
> ---
>
> ## Q1: the paper relies too heavily on labeled force-video datasets, rather than unlabeled datasets
>
> We consider the **primary contribution of our paper to be:** the definition of the task (adding physical forces as a conditioning signal for video generation models) and the initial demonstration that this task is not only feasible but yields interesting and potentially promising results. While there may be other different approaches or datasets that we could have used to train the model, we believe that this shows that there are many potentially interesting and exciting follow ups to our proposed task.
>
> We would also note that:
> 1. Our method is enabled by the robust motion priors present in large, **unlabeled** (with respect to physical forces) video datasets used to train the base model. This allows us to effectively guide the model with a limited amount of synthetic force-video data.
>
> 2. To the best of our knowledge, there is **no** existing fine-grained force-video dataset. One of our contributions is to provide such a dataset, and the best practices on how to curate it. Our data is synthesized with physics simulators, thus can be scaled up if necessary, without any manual annotations.
> 3. We agree that using unlabeled video data to train force conditioned models (e.g. by inferring the forces from those videos with a machine learning algorithm) is a promising approach to generalize the force conditioning to even more diverse scenarios. However, to the best of our knowledge such algorithms are still nascent.
>
> ---
>
> ## Q2: defining HDRI
>
> HDRI stands for “High Dynamic Range Image”. It is a media format that is useful when you need a background image for composited scenes, as it stores a much wider range of luminance values than regular images. We’ll add more information about this to the next draft, thanks for the suggestion.
>
> ---
>
> ## Q3: why was Blender chosen over other platforms or more advanced physics simulators?
>
> While we agree that advanced engines like Unity and Unreal Engine 5 could have been beneficial, their licensing terms explicitly prohibit their use for training generative AI models. We therefore chose Blender for its highly permissive license. We also believe that using a more advanced simulation engine would indeed improve the performance of our method.
>
> ---
>
> ## Q4: does the method face bottlenecks relating to data diversity or label quality?
>
> We investigated the impact of data diversity on model performance, and concluded that:
>
> 1. More diverse synthetic data (more background, more flags) indeed helps generalization.
> 2. Our method is nonetheless robust and generalizes to diverse scenarios.
>
> These results of these ablation studies are summarized in Section 5, and detailed figures are in Figure 7 in the appendix.
>
> ---
>
> ## Q5: why was PhysDreamer not included as a baseline for comparison in Table 1?
>
> Unlike our method, **PhysDreamer** is not a general-purpose model; it must be fit to each individual scene. To use it, you first need to learn a 3D Gaussian representation of the scene and estimate physical properties like mass, Young’s modulus, and Poisson's ratio for each object. This approach is not suitable for many objects in our benchmark, such as a train moving on a track, where for example, Young's modulus does not make sense as a modeling choice.
>
> In contrast, our **Force Prompting** method is designed for a general task. To apply it, you simply upload an image of the scene, and our neural architecture intuitively models the object's physics.
>
> We did, however, conduct a direct comparison against PhysDreamer in **Table 2**, where we ran our Force Prompting model on six of PhysDreamer's flower demos.
>
> ---
>
> ## Q6: more details about comparison with PhysDreamer
>
> We believe that the evaluation metrics and testing scenarios were consistent. We will detail here the experimental design choices that we made to ensure a reasonable comparison between PhysDreamer and Force Prompting. We’ll add these details to the next draft of the paper.
>
> * We took 6 flower demo videos from the PhysDreamer teasers (Alocacia, Carnation, Orange Rose, Red Rose, White Rose, Tulip) and screenshotted a still frame from the video that did not have any force annotation on it.
> * We passed these six frames, as well as the 6 equivalent force prompts (i.e. the same force vector from the original demo) into the Force Prompting model.
> * We presented these videos side by side in the human study, which we served using Qualtrics. Both videos for a given scene were shown to the human annotator simultaneously, with left/right randomization.
>
> The purpose of comparing Force Prompting to PhysDreamer was to demonstrate that Force Prompting can successfully generalize to various plants (Alocacia, Orange Rose, Red Rose, White Rose, Tulip), without training on them. In contrast, the PhysDreamer method requires a learned 3D Gaussian representation, as well as estimates for the mass, Young’s modulus, and Poisson ratio for the object. To be clear, we’re not claiming that Force Prompting outperforms PhysDreamer on any sort of motion/force/video quality metrics. Rather, we conducted that human study just to demonstrate that our Force Prompting method, which doesn't use any physics simulator or 3D assets at inference time, is *somewhat comparable* to PhysDreamer, which does require those inductive biases and explicit representations.
>
> ---
>
> ## Q7: why was PhysGen not included as a baseline for comparison?
>
> PhysGen is specifically designed for rigid body simulation. As a result it would not work on approximately 78% of the scenes in our benchmark (e.g. the complex and oscillatory motions– flowers, ornaments, apple trees). As for the remaining 22% of the scenes in our benchmark (e.g. rolling toy cars), setting up PhysGen to run inference is non-trivial, requiring:
>
> 1. A “perception” data preprocessing step, which involves extracting a foreground segmentation mask, synthesizing an inpainted background image, extracting normal, shading, and albedo maps, and physics properties of foreground objects
> 2. Specifying the simulation parameters (force on each object, gravity, initial velocity and acceleration)
>
> In contrast, with our Force Prompting model you only need to provide the first frame and a force vector. We wanted to include a comparison of our benchmark against another method, but to the best of our knowledge, the only system that enables force control in similarly diverse settings is WonderPlay (which integrates physical simulators, unlike our work), which was made public after the NeurIPS deadline.
>
> ---
>
> ## Q8: hyperlinks broken
>
> We’re sorry about that! We have identified a fix for this and it will be solved in the next draft of the paper.
>
> ---
>
> ## Q9: clarify the choice of terminology “localized point force” and “global force fields”
>
> We categorize simulator-based forces into two distinct types: localized forces that are applied to a specific object in a scene (e.g. pushes and pulls), and global forces that affect the entire scene uniformly (e.g. wind). We built one model for each case. We termed the former a “localized point force model”, and the latter a “global wind force”. We think that if we clarify this at the start of the paper that it would make the definitions more clear.

---

> ### Author Response · Authors · 2025-08-03
>
> We wanted to follow up to make sure that your concerns are being properly addressed. Please let us know if there are additional questions that we can provide further information or experiments on. Thank you again for your time and effort reviewing our paper!

---

> > ### Comment · Reviewer_ugp2 · 2025-08-06
> >
> > After carefully reading the authors’ rebuttal and follow-up clarifications, I am now more confident in the contributions of this work. The authors have thoroughly addressed my major concerns, especially those related to dataset construction, baseline comparisons, and the rationale behind the choice of simulation tools.
> >
> > I suggest the authors include in their revised manuscript the reasoning provided on why PhysDreamer and PhysGen were not ideal baselines for certain evaluations. I also understand that WonderPlay was released after the NeurIPS submission deadline, but it would be helpful to briefly acknowledge it in the revised version for completeness.
> >
> > One area I found a bit unclear in the original submission was the choice of terminology—specifically, “localized point force” and “global force fields.” It would improve clarity if the paper could offer a more explicit definition and justification for these terms earlier on.
> >
> > Despite these minor issues, I believe the novelty, feasibility, and potential impact of this paper clearly outweigh the concerns. Therefore, I raise my rating to 5: Accept.

---

> > ### Comment · Area_Chair_kk9E · 2025-08-06
> > **Final comment**
> >
> > Dear Reviewer,
> > Could you please read the rebuttal, update your final review, and share any remaining follow-up questions, if any? Also, kindly acknowledge once this is done. Thank you!

---

### Official Review · Reviewer_fCVZ · 2025-07-05

**Clarity:** 3
**Significance:** 2
**Originality:** 2
**Rating:** 4
**Confidence:** 4

**Summary:**

This article introduces physical forces as novel conditioning signals for video generation models. The core insight is that pretrained video models possess latent intuitive physics priors that can be elicited via fine-tuning on limited synthetic data. There are several innovations. Physics-based control encoding, Sim2Real generalization and emergent mass understanding. Human evaluations demonstrate superior force adherence and realism over text-conditioned baselines and trajectory-based methods.

**Questions:**

Please refer to weakness.

**Ethical Concerns:**

["NO or VERY MINOR ethics concerns only"]

**Final Justification:**

The rebuttal addressed my concerns, I will raise the rating to borderline accept.

**Limitations:**

Please refer to weakness.

**Quality:**

3

**Strengths And Weaknesses:**

Strength

1. The authors carried out a novel and interesting task, and it achieved certain results. Additionally, they provide codes in supplementary materials.

2. The experimental results are very abundant, with a wide variety of objects, and the experimental data and network parameters descriptions are also very detailed.

3. Human evaluation shown in Table 1 reflects the effectiveness of proposed model.

Weakness

1. In some cases, such as in the example of a scene where the wind blows, usually only the target object moves while the background remains stationary. This does not conform to the principles of physics.

2. Figure 3 is hard to follow. What do the blue lines in the last two columns mean?

3. The presented model has poor performance on real physics, such as fluid dynamics. As shown in Figure 4, the diffusion of smoke affected by the wind is very unrealistic. The smoke emerging from the cigarette end cannot be straight, and the inertia is not so significant.

---

> ### Author Rebuttal · Authors · 2025-07-30
>
> Thank you for the thoughtful feedback and the recognition of our "novel and interesting tasks" with "abundant" experimental results. We consider the **primary contribution of our paper to be:** the definition of the task (adding physical forces as a conditioning signal for video generation models) and the initial demonstration that this task is not only feasible but yields interesting and potentially promising results. In particular, we show that despite limited training data, the model can generalize the physical conditioning signals to many different images and scenarios.
>
> We agree with your assessment that our model, based on CogVideoX, sometimes produces video outputs that deviate from real-world physics, especially in complex scenarios like fluid dynamics. However, we believe our approach remains highly promising due to the significant advancements in video generation quality over the last two years. State-of-the-art models, such as Sora and Veo 3, have achieved incredible motion and physics realism, approaching photo-realism.
> Because these cutting-edge models are proprietary, we opted to use an open-source model that provided a reasonable balance between quality and computational feasibility. We expect that applying our proposed techniques to a more capable, state-of-the-art base model would yield substantially better results. This is partially supported by our current findings: for certain scenarios like wind blowing on cloth and leaves, or objects oscillating after being poked, the CogVideoX-based model already produces more physically accurate results. We are optimistic that as the underlying video generation models continue to improve, our method's performance across a wider range of physical scenarios will also improve significantly.
>
> Below we provide detailed answers to your questions.
>
> ---
>
> ## Q1: sometimes the target moves while the background remains stationary
> We agree that the video generation model can violate physical principles, especially for smaller objects. This is a known limitation of earlier video generation models and an active area of research (e.g. motion-based losses) for video generation base models.
> To improve it in our setting, we adapted our training data during post-training. As shown in Appendix Figure 7, we found that adding more flags and increasing background diversity significantly mitigates these errors. While not completely solved, our experiments suggest that enhancing the quality and diversity of our synthetic training data is a promising direction for future work.
>
> ---
>
> ## Q2: clarifying the blue lines in Figure 3
>
> The blue lines visualize a time-lapse of the object's movement, with each line segment connecting its position from one frame to the next. For the flower, the sequence unfolds like this:
>
> - *Frame 1:* A single blue dot marks the initial position of the point force target.
> - *Frame 2:* A blue arc shows the flower's path as it moves downwards from its previous location.
> - *Frame 3:* Another blue arc traces the flower's movement as it arcs back upwards.
>
> We will update the figure's caption in the revised version to more clearly illustrate this time-lapse.
>
> ---
>
> ## Q3: unrealistic videos with fluid dynamics
>
> As previously discussed, we acknowledge that some samples demonstrate a limited understanding of physics. However, we believe this paper introduces a novel task and approach whose performance will improve significantly as the underlying video generation models advance. Our method's effectiveness is inherently tied to the physics quality of its base model. For example, while our wind force model performs well with objects like cloth and leaves, its performance is limited in scenarios where the base model itself struggles with physics realism.
>
> We provide evidence for this relationship in our human study. As shown in **Table 1**, the physics quality of our Force Prompting method is directly correlated with the physics quality of the base model. This suggests that with a more capable foundation model, our approach can achieve substantially better results.

---

> ### Author Response · Authors · 2025-08-03
>
> We wanted to follow up to make sure that your concerns are being properly addressed. Please let us know if there are additional questions that we can provide further information or experiments on. Thank you again for your time and effort reviewing our paper!

---

> > ### Comment · Reviewer_fCVZ · 2025-08-06
> >
> > The rebuttal addressed my concerns, I will raise the rating to borderline accept.

---

### Note · Authors · 2025-08-14

Dear Reviewers + ACs,

We would like to sincerely thank the reviewers for their constructive feedback, as well as the ACs for their support. We also wanted to thank reviewers fCVZ and ugp2 for raising their ratings.

We summarize the paper’s primary contributions:

1. **A New Control Modality:** We introduce physical forces as conditioning signals for video generation, proposing models for localized point forces and global wind effects. The models do not use a physics simulator or any 3D assets at inference time.
2. **Sim2Real Generalization:** We demonstrate that a model fine-tuned on minimal synthetic data (\~16K videos) with modest computational resources (\~one day on four A100 GPUs) can achieve broad, robust generalization, applying learned physical dynamics to a wide variety of unseen objects, materials, and scenes.
3. **Blueprint for Controllable Generalization:** Through extensive ablations, we identify the key ingredients for successful generalization: visual diversity of synthetic training data, and keyword-driven prompting.
4. **Emergent Physical Intuition:** We show that the model acquires an implicit understanding of physics, such as recognizing that lighter objects should travel farther than heavier ones under the same force. This understanding is robust enough to persist even in complex, multi-object scenarios.
5. **Zero-Shot Compositionality:** The model can interpret and render multiple, simultaneous force prompts without ever having seen such examples during training.
6. **Open Resources:** We commit to releasing all of our code, model weights, and synthetic datasets to the community.

We are grateful for all reviewers’ constructive feedback, which have helped us strengthen our work. We will incorporate the changes into the revised version. Thank you!

Sincerely,

Force Prompting authors

---

### Decision · Program_Chairs · 2025-09-17

**Decision:**

Accept (poster)

**Comment:**

This paper proposes a novel image-to-video generation framework that leverages physical forces as conditioning signals. The method demonstrates realistic physics under wind and point-force conditioning and, notably, achieves strong generalization through careful conditioning design and fine-tuning with a small amount of simulation data. Post-rebuttal, all reviewers recognized the solid results, emergent properties, and potential impact of this work in advancing physically grounded video generation. The AC concurs and recommends acceptance.